# The transcription factor EPAS1 links DOCK8 deficiency to atopic skin inflammation via IL-31 induction

Kazuhiko Yamamura[1,2], Takehito Uruno[1,3], Akira Shiraishi[1], Yoshihiko Tanaka[4], Miho Ushijima[1], Takeshi Nakahara[2], Mayuki Watanabe[1], Makiko Kido-Nakahara[2], Ikuya Tsuge[5], Masutaka Furue[2] & Yoshinori Fukui[1,3]

Mutations of DOCK8 in humans cause a combined immunodeficiency characterized by atopic dermatitis with high serum IgE levels. However, the molecular link between DOCK8 deficiency and atopic skin inflammation is unknown. Here we show that CD4$^+$ T cells from DOCK8-deficient mice produce large amounts of IL-31, a major pruritogen associated with atopic dermatitis. IL-31 induction critically depends on the transcription factor EPAS1, and its conditional deletion in CD4$^+$ T cells abrogates skin disease development in DOCK8-deficient mice. Although EPAS1 is known to form a complex with aryl hydrocarbon receptor nuclear translocator (ARNT) and control hypoxic responses, EPAS1-mediated Il31 promoter activation is independent of ARNT, but in collaboration with SP1. On the other hand, we find that DOCK8 is an adaptor and negative regulator of nuclear translocation of EPAS1. Thus, EPAS1 links DOCK8 deficiency to atopic skin inflammation via IL-31 induction in CD4$^+$ T cells.

[1] Division of Immunogenetics, Department of Immunobiology and Neuroscience, Medical Institute of Bioregulation, Kyushu University, 3-1-1 Maidashi, Higashi-ku, Fukuoka 812-8582, Japan. [2] Department of Dermatology, Graduate School of Medical Sciences, Kyushu University, 3-1-1 Maidashi, Higashi-ku, Fukuoka 812-8582, Japan. [3] Research Centre for Advanced Immunology, Kyushu University, 3-1-1 Maidashi, Higashi-ku, Fukuoka 812-8582, Japan. [4] Department of Functional Bioscience, Section of Infection Biology, Fukuoka Dental College, 2-15-1 Tamura, Sawara-ku, Fukuoka 814-0175, Japan. [5] Department of Pediatrics, School of Medicine, Fujita Health University, 1-98 Dengakugakubo, Kutsukake-cho, Toyoake 470-1192, Japan. Correspondence and requests for materials should be addressed to Y.F. (email: fukui@bioreg.kyushu-u.ac.jp).

Homozygous and compound heterozygous mutations in *DOCK8* cause a combined immunodeficiency characterized by recurrent viral infections, early onset malignancy and atopic dermatitis (AD)[1–5]. As patients with *DOCK8* mutations have elevated serum IgE levels, this disorder has been classified as a form of autosomal recessive hyper IgE syndrome (HIES)[4,5]. DOCK8 is an evolutionarily conserved guanine nucleotide exchange factor (GEF) for Cdc42 (ref. 6). Accumulating evidence indicates that human patients with *DOCK8* mutations have morphological and functional abnormalities of leukocytes[1,7–9]. In addition, the important roles of DOCK8 in leukocytes have been demonstrated using animal models. For example, *N*-ethyl-*N*-nitrosourea-mediated mutagenesis in mice has shown that DOCK8 regulates immunological synapse formation in B cells, and is required for development or survival of memory CD8[+] T cells, natural killer T cells and type 3 innate lymphoid cells[9–14]. On the other hand, by generating DOCK8-deficient (*Dock8*[−/−]) mice, we and others have revealed that DOCK8 is essential for interstitial migration of dendritic cells[6,15]. However, the molecular link between DOCK8 deficiency and atopic skin inflammation is not defined.

Interleukin 31 (IL-31) is a recently discovered cytokine that is related to the IL-6 cytokine family in terms of its structure and receptor complex[16,17]. IL-31 is expressed by activated T cells and is preferentially produced by T helper type 2 (Th2) CD4[+] T cells. IL-31 signals via a heterodimeric receptor composed of IL-31 receptor A (IL31RA; also known as GPL or GLM-R) and oncostatin M receptor (OSMR)[16,17], both of which are expressed in various cell-types, including neurons of dorsal root ganglia[17–19]. Mice treated with intradermal injection of IL-31 and transgenic mice over-expressing IL-31 exhibit scratching behaviour and develop severe dermatitis[16,19]. In addition, in patients with AD, IL-31 is expressed by cutaneous lymphocyte antigen-positive skin homing CD4[+] T cells[18,20], and serum IL-31 levels have been shown to correlate with the disease activity[21]. Moreover, a phase 1 clinical trial has proven the apparent anti-pruritic effect of IL-31 receptor antibody in AD[22]. Thus, IL-31 is a T-cell derived cytokine closely associated with pruritus in AD patients.

Endothelial PAS domain protein 1 (EPAS1) is a member of basic-helix-loop-helix/PAS domain containing transcription factors, and has a high homology to hypoxia-inducible factor (HIF)-1α, thus termed HIF-2α (ref. 23). Like HIF-1α, EPAS1 forms a complex with aryl hydrocarbon receptor nuclear translocator (ARNT; also know as HIF-1β) and transactivates target genes to respond to environmental stressors such as hypoxia[24–26]. Although the role of EPAS1 in the immune system is unexplored, gene network analysis suggests that EPAS1 is a direct target of signal transducer and activator of transcription-6 (STAT6) and acts as a hub protein in IL-4-mediated transcription circuitries in human CD4[+] T cells[27,28]. Therefore, EPAS1 may have a specific role in differentiation and effector functions of Th2 cells. In this study, we identify EPAS1 as an important transcription factor for IL-31 induction in CD4[+] T cells. Interestingly, EPAS1 mediates *Il31* promoter activation independently of ARNT, but in collaboration with the transcription factor SP1. On the other hand, we find that DOCK8 negatively regulates nuclear translocation of EPAS1 and its deletion leads to severe skin disease in a mouse model. Our results thus indicate that EPAS1 links DOCK8 deficiency to atopic skin inflammation via IL-31 induction in CD4[+] T cells.

## Results

**Production of IL-31 by *Dock8*[−/−] CD4[+] T cells.** To examine the role of DOCK8 in antigen-specific T cell responses, we developed *Dock8*[−/−] mice expressing OTII T-cell receptor (TCR) that recognizes OVA peptide (OVA323–339) presented by I-A[b] molecules on the genetic background of C57BL/6 mice (designated *Dock8*[−/−] OTII Tg mice). Flow cytometric analyses revealed that T cell development in the thymus occurred normally even in the absence of DOCK8 (Fig. 1a). Although the proportion and the number of T cells in the spleen were reduced in *Dock8*[−/−] OTII Tg mice (Fig. 1a,b), as reported in *Dock8*[−/−] non-TCR Tg mice[6], the majority of peripheral T cells were CD4[+] T cells expressing Vα2[+]Vβ5[+] OTII TCR, irrespective of DOCK8 expression (Fig. 1a). These CD4[+] T cells proliferated comparably when stimulated with OVA peptide in the presence of I-A[b]-expressing C57BL/6 spleen cells (Fig. 1c). Interestingly, however, we found that the level of *Il31* transcript markedly increased in CD4[+] T cells from *Dock8*[−/−] OTII Tg mice 24 h after stimulation, as compared with that in *Dock8*[+/−] OTII Tg CD4[+] T cells (Fig. 1d). In contrast, *Il2* and *Il4* gene expressions in stimulated CD4[+] T cells were unchanged between *Dock8*[+/−] and *Dock8*[−/−] OTII Tg mice (Fig. 1d). This effect of DOCK8 deficiency on *Il31* induction became more evident when activated CD4[+] T cells were recovered 96 h after antigen stimulation, and re-stimulated with anti-CD3ε antibody. Indeed, compared with the level of unstimulated control samples, *Il31* transcript increased 4,369-fold in *Dock8*[−/−] OTII Tg CD4[+] T cells at 3 h after secondary stimulation, and this value was 49.0 times higher than that of *Dock8*[+/−] OTII Tg CD4[+] T cells (Fig. 1e). Consistent with this finding, ELISA revealed that CD4[+] T cells from *Dock8*[−/−] OTII Tg mice produced larger amounts of IL-31 on TCR stimulation (Fig. 1f). These results indicate that DOCK8 negatively regulates antigen-induced IL-31 production by CD4[+] T cells.

Although IL-31 has been implicated in pruritus in AD, *Dock8*[−/−] OTII Tg mice showed neither skin inflammation nor IgE elevation (Supplementary Fig. 1). To assess whether CD4[+] T cells from *Dock8*[−/−] OTII Tg mice have the potential to induce itch, we adoptively transferred them into mice ubiquitously expressing OVA under the control of the cytomegalovirus immediate early enhancer-chicken β-actin hybrid promoter (designated CAG-OVA mice). As a result, antigen-stimulated CD4[+] T cells from *Dock8*[−/−], but not *Dock8*[+/−], OTII Tg mice induced scratching behaviour in CAG-OVA mice at 5–7 h after transfer (Fig. 1g). These results indicate that *Dock8*[−/−] CD4[+] T cells are capable of inducing itch when they are activated *in vivo*.

**Atopic skin inflammation in *Dock8*[−/−] AND Tg mice.** To examine whether the effect of DOCK8 deficiency could be extended to other CD4[+] T cells with different antigen specificity, we developed *Dock8*[−/−] mice expressing AND TCR on the genetic background of C57BL/6 mice (designated *Dock8*[−/−] AND Tg mice). AND TCR is a product of artificial αβ-chain combination, which recognizes moth cytochrome C peptide (MCC88–103) in the context of I-E[k] molecules; yet, it is known that CD4[+]CD8[+] thymocytes expressing AND are also selected to mature in the presence of I-A[b] molecules[29]. As seen in *Dock8*[−/−] OTII Tg mice, CD4[+] T cells normally developed in *Dock8*[−/−] AND Tg mice and responded to MCC peptide in the presence of I-E[k]-expressing B10.BR spleen cells (Supplementary Fig. 2a–c). In addition, CD4[+] T cells from *Dock8*[−/−] AND Tg mice produced large amounts of IL-31 on secondary stimulation (Supplementary Fig. 2d). Thus, DOCK8

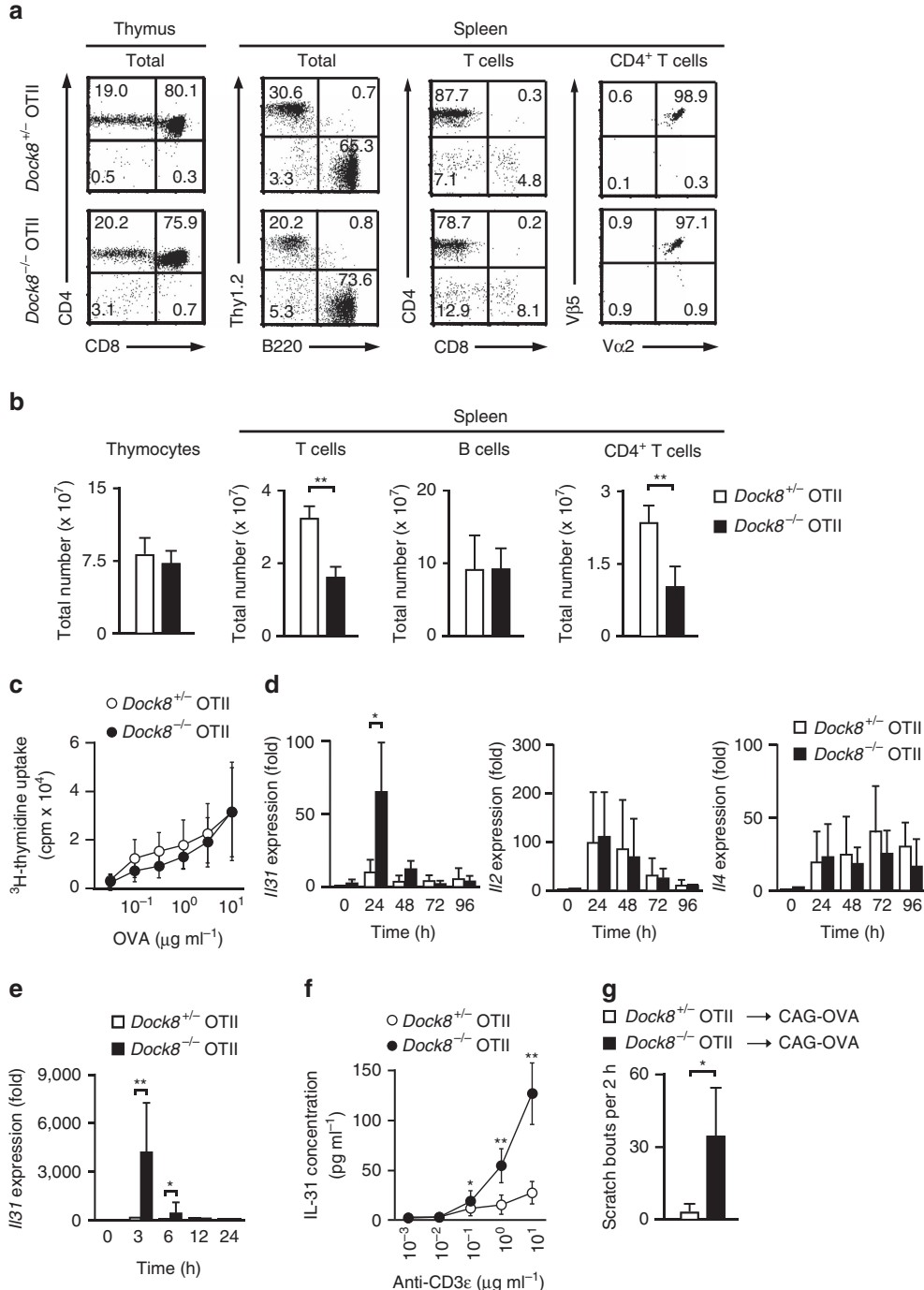

**Figure 1 | DOCK8 negatively regulates IL-31 induction in CD4$^+$ T cells. (a,b)** Flow cytometric analyses of thymocytes and spleen cells from 6- to 8-week-old $Dock8^{+/-}$ and $Dock8^{-/-}$ OTII Tg mice. The numerals in quadrants indicate the percentage of each subset of leukocytes. Data are expressed as mean ± s.d. of 5 mice per group. **$P < 0.01$ (two-tailed Student's $t$-test). See Supplementary Fig. 10 for FACS gating strategy. **(c)** Antigen-specific proliferation of CD4$^+$ T cells from $Dock8^{+/-}$ and $Dock8^{-/-}$ OTII Tg mice. Data are expressed as mean ± s.d. of 12 samples per group. **(d)** Induction of cytokine gene expression in CD4$^+$ T cells from $Dock8^{+/-}$ and $Dock8^{-/-}$ OTII Tg mice after primary stimulation with OVA peptide. Expression (fold increase) is relative to that of unstimulated $Dock8^{+/-}$ samples. Data are expressed as mean ± s.d. of 8 samples per group. *$P < 0.05$ (two-tailed Student's $t$-test). **(e)** $Il31$ gene expression in CD4$^+$ T cells from $Dock8^{+/-}$ and $Dock8^{-/-}$ OTII Tg mice after secondary stimulation with anti-CD3ε and anti-CD28 antibodies. Expression (fold increase) is relative to that of $Dock8^{+/-}$ samples without secondary stimulation. Data are expressed as mean ± s.d. of 10 samples per group. *$P < 0.05$; **$P < 0.01$ (two-tailed Student's $t$-test). **(f)** ELISA showing increased IL-31 production by $Dock8^{-/-}$ OTII Tg CD4$^+$ T cells after secondary stimulation. Data are mean ± s.d. of 9 samples per group. *$P < 0.05$; **$P < 0.01$ (two-tailed Student's $t$-test). **(g)** Induction of itch in CAG-OVA mice by adoptive transfer of activated CD4$^+$ T cells from $Dock8^{-/-}$, but not $Dock8^{+/-}$, OTII Tg mice. Data are expressed as mean ± s.d. of four mice per group. *$P < 0.05$ (two-tailed Mann–Whitney test).

generally acts as a negative regulator for IL-31 induction in CD4$^+$ T cells, irrespective of their antigen specificity.

Surprisingly, we found that $Dock8^{-/-}$ AND Tg mice developed severe skin inflammation by the age of 14–15 weeks. The severity of dermatitis was grossly assessed by the SCORAD (human SCORing Atopic Dermatitis) that was modified for the use in mice[30,31]. Irrespective of the sex of mice, skin phenotype first appeared 7–8 weeks after birth, generally worsened with the age, and was never observed in $Dock8^{+/-}$ AND Tg littermate mice (Fig. 2a). Consistent

with this, the proportion of CD44$^+$CD62L$^-$ activated CD4$^+$ T cells gradually increased in $Dock8^{-/-}$ AND Tg mice with the age (Fig. 2b). In addition, the serum concentrations of IgE, but not IgG2b, increased in $Dock8^{-/-}$ AND Tg mice at 12 and 18 weeks old (Fig. 2c). $Dock8^{-/-}$ AND Tg mice exhibited intense scratching behaviour at 12 and 18 weeks old (Fig. 2d), indicating that this skin inflammation was pruritic. Histological and immunohistochemical analyses of the skin from $Dock8^{-/-}$ AND Tg mice revealed acanthosis, hyperkeratosis and mild spongiosis with massive infiltration of CD4$^+$ T cells

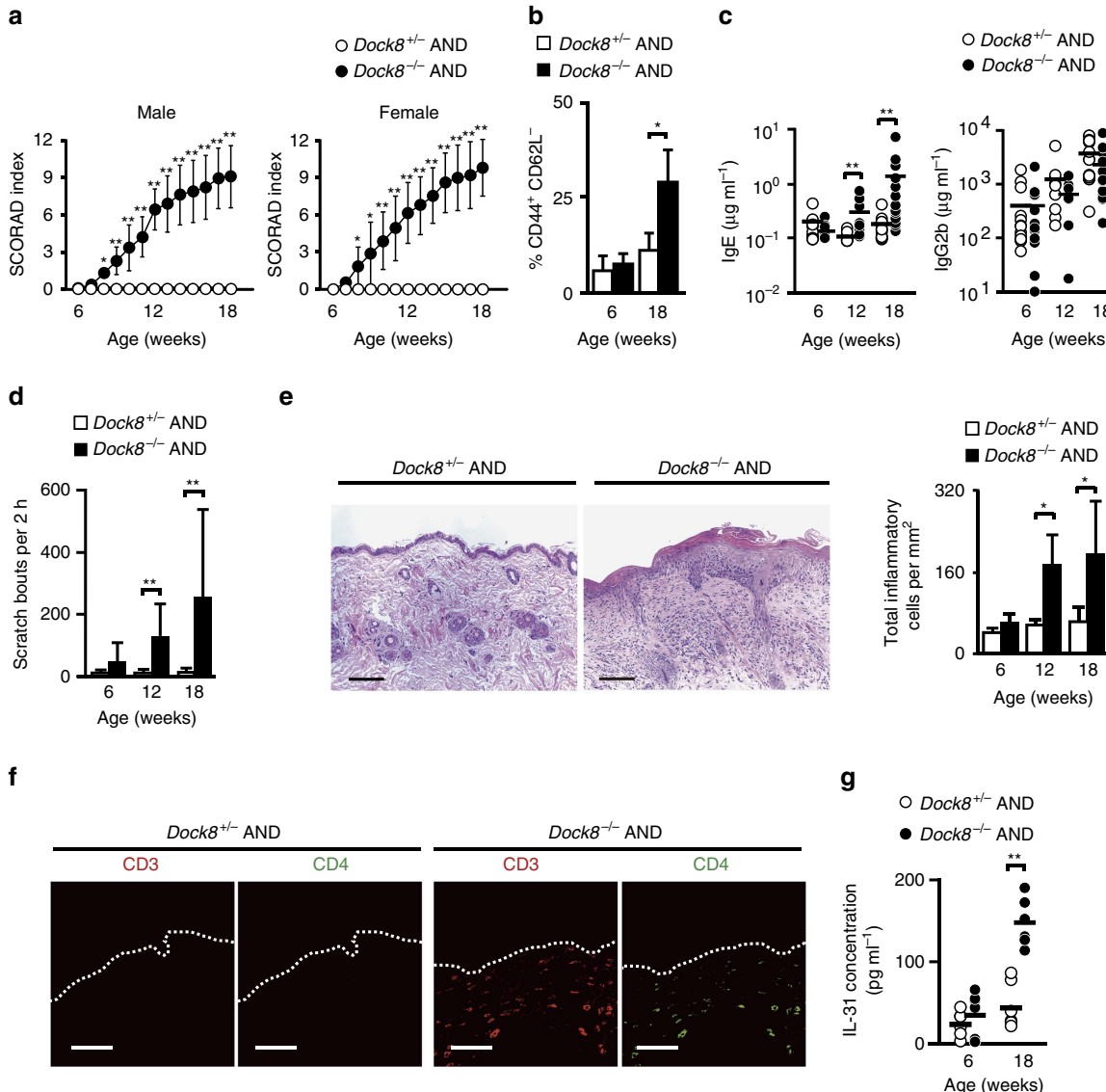

**Figure 2 | Development of atopic skin inflammation in $Dock8^{-/-}$ AND Tg mice.** (**a**) SCORAD index of $Dock8^{+/-}$ (male, $n = 20$; female, $n = 15$) and $Dock8^{-/-}$ (male, $n = 20$; female, $n = 15$) AND Tg littermates. Data are expressed as mean ± s.d. *$P < 0.05$; **$P < 0.01$ (two-tailed Student's $t$-test). (**b**) The percentage of CD44$^+$CD62L$^-$ activated CD4$^+$ T cells in total PLN CD4$^+$ T cells from $Dock8^{+/-}$ and $Dock8^{-/-}$ AND Tg littermates at 6 and 18 weeks old. Data are expressed as mean ± s.d. of 4 mice per group. *$P < 0.05$ (two-tailed Mann–Whitney test). (**c**) Serum concentrations of IgE and IgG2b in $Dock8^{+/-}$ and $Dock8^{-/-}$ AND Tg mice at 6 ($n = 13$), 12 ($n = 8$ or 9) and 18 ($n = 14$) weeks old. The lines indicate the mean values. **$P < 0.01$ (two-tailed Mann–Whitney test). (**d**) Comparison of scratching bouts between $Dock8^{+/-}$ and $Dock8^{-/-}$ AND Tg mice at 6 ($n = 8$), 12 ($n = 6$) and 18 ($n = 8$) weeks old. Data are expressed as mean ± s.d. **$P < 0.01$ (two-tailed Mann–Whitney test). (**e**) Haematoxylin and eosin staining of the skin from 18-week-old littermates. Scale bars, 100 μm. The number of inflammatory cells per mm$^2$ was compared between $Dock8^{+/-}$ and $Dock8^{-/-}$ AND Tg mice. In each experiment, three fields were examined per mouse. Data are expressed as mean ± s.d. of four mice group. *$P < 0.05$ (two-tailed Mann–Whitney test). (**f**) Immunofluorescence staining showing infiltration of CD4$^+$ T cells in the skin of $Dock8^{-/-}$ AND Tg mice. Scale bars, 50 μm. (**g**) Serum concentrations of IL-31 in $Dock8^{+/-}$ and $Dock8^{-/-}$ AND Tg mice at 6 ($n = 6$) and 18 ($n = 6$) weeks old. The lines indicate the mean values. **$P < 0.01$ (two-tailed Student's $t$-test).

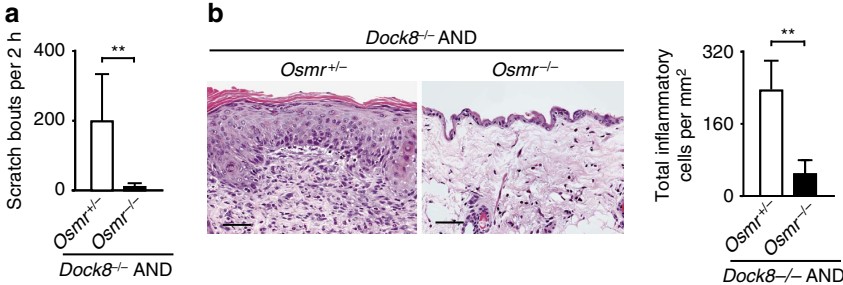

**Figure 3 | OSMR-dependent skin disease development in $Dock8^{-/-}$ AND Tg mice.** (**a**) Comparison of scratching bouts between $Osmr^{+/-}$ and $Osmr^{-/-}$ $Dock8^{-/-}$ AND Tg mice at 18 weeks old. Data are expressed as mean ± s.d. of 6 mice per group. **$P < 0.01$ (two-tailed Student's $t$-test). (**b**) Haematoxylin and eosin staining of the skin from 18-week-old littermates. Scale bars, 50 μm. The number of inflammatory cells per mm² was compared between $Osmr^{+/-}$ and $Osmr^{-/-}$ $Dock8^{-/-}$ AND Tg mice at 18 weeks old. In each experiment, 3 fields were examined per mouse. Data are expressed as mean ± s.d. of 5 mice per group. **$P < 0.01$ (two-tailed Mann–Whitney test).

and, to lesser extent, eosinophils (Fig. 2e,f and Supplementary Fig. 3). The skin-infiltrating CD4$^+$ T cells in $Dock8^{-/-}$ AND Tg mice mainly expressed Th2 cytokines including $Il31$ (Supplementary Fig. 4). More importantly, $Dock8^{-/-}$ AND Tg mice showed significant increase in serum IL-31 levels at 18 weeks, compared with those of $Dock8^{+/-}$ AND Tg mice (Fig. 2g).

IL-31 transmits the signals via a heterodimeric receptor composed of IL31RA and OSMR[16,17]. To examine whether IL-31-mediated signals are involved in the disease development, we deleted the expression of OSMR in $Dock8^{-/-}$ AND Tg mice by crossing them with OSMR-deficient mice ($Osmr^{-/-}$)[32]. OSMR deficiency did not affect serum IL-31 levels (Supplementary Fig. 5). However, in AND Tg mice lacking both DOCK8 and OSMR ($Dock8^{-/-}$ $Osmr^{-/-}$ AND Tg), scratching bouts per 2 h were reduced to 11.5 ± 9.3, as compared with 200.8 ± 135.8 in $Dock8^{-/-}$ $Osmr^{+/-}$ AND Tg mice (Fig. 3a), and they did not develop skin inflammation (Fig. 3b). So far, no evidence has been provided that oncostatin M is involved in itch induction. Therefore, these results suggest that $Dock8^{-/-}$ AND Tg mice spontaneously develop atopic skin inflammation through the mechanism dependent on IL-31 signalling.

**Identification of EPAS1 as a regulator for IL-31 induction**. To explore the underlying mechanism of IL-31 overproduction by $Dock8^{-/-}$ CD4$^+$ T cells, we first performed microarray analysis, and found that 856 genes were expressed at higher levels in $Dock8^{-/-}$ AND Tg CD4$^+$ T cells than $Dock8^{+/-}$ AND Tg controls after antigen stimulation. These included 40 genes encoding putative transcriptional factors (Supplementary Table 1), one of which was EPAS1. When pMX vector encoding EPAS1-IRES-GFP was expressed in WT CD4$^+$ T cells by retroviral transfer, TCR stimulation-induced $Il31$ gene expression was significantly augmented, compared with that of the control expressing GFP alone (Fig. 4a). On the other hand, induction of $Il31$ gene expression in $Dock8^{-/-}$ AND CD4$^+$ T cells was markedly suppressed by knocking down $Epas1$ gene expression using siRNA (Fig. 4b). Similar results were obtained when $Il31$ gene expression was analysed in CD4$^+$ T cells from $Dock8^{-/-}$ AND Tg mice lacking EPAS1 expression in a CD4$^+$ T cell-specific manner (CD4-Cre$^+$ $Epas1^{lox/lox}Dock8^{-/-}$ AND Tg mice) (Fig. 4c). More importantly, scratching behaviour and skin disease development as well as an increased serum IL-31 level were cancelled in all CD4-Cre$^+$ $Epas1^{lox/lox}Dock8^{-/-}$ AND Tg mice tested (Fig. 4d–f). Thus, EPAS1 functions as a master regulator for IL-31 induction in CD4$^+$ T cells and is

required for development of atopic skin inflammation in $Dock8^{-/-}$ AND Tg mice.

**EPAS1 acts with SP1 in $Il31$ promoter activation**. Having identified EPAS1 as a key molecule for $Il31$ induction, we next examined how EPAS1 acts in $Il31$ promoter activation. For this purpose, we created reporter construct containing $Il31$ promoter sequence ($-1,367$ to $-1$) and the luciferase gene. When this reporter construct was expressed in mouse embryonic fibroblasts (MEFs), $Il31$ promoter activation was induced in the presence of WT EPAS1 (Fig. 5a). However, the expression of EPAS1 mutants lacking N-terminal or C-terminal activation domain (ΔN-TAD and ΔC-TAD) failed to induce promoter activation (Fig. 5a). EPAS1 is known to form a complex with ARNT via PAS-B domain[26]. Surprisingly, however, deletion of neither PAS-B nor bHLH did affect $Il31$ promoter activation (Fig. 5a). Consistent with this, EPAS1-mediated $Il31$ promoter activation occurred normally even when $Arnt$ gene was knocked down (Fig. 5b). By deleting $Il31$ promoter region, we identified the critical region for EPAS1-mediated transactivation (Fig. 5c), which included a consensus SP1-binding sequence GGCC at the position from $-118$ to $-115$ (Fig. 5d). Indeed, electrophoretic mobility shift assay (EMSA) showed that SP1 bound to this sequence (Fig. 5e). When this GGCC sequence was mutated to AAGT, EPAS1-mediated $Il31$ promoter activation was almost completely lost (Fig. 5d). More importantly, EPAS1-mediated $Il31$ promoter activation significantly diminished by knocking down $Sp1$ gene expression (Fig. 5b). In addition, chromatin immunoprecipitation (ChIP) assay revealed that SP1 was recruited to $Il31$ promoter in activated CD4$^+$ T cells from $Dock8^{-/-}$ AND Tg mice and its recruitment was significantly reduced in the absence of EPAS1 (Fig. 5f). Collectively, these results indicate that EPAS1 induces $Il31$ promoter activation, independently of ARNT, but in collaboration with SP1.

**DOCK8 negatively regulates nuclear translocation of EPAS1**. EPAS1 mediates promoter activation of target genes in the nucleus after translocation from the cytoplasm. As DOCK8 is also expressed in MEFs (Supplementary Fig. 6), we prepared WT and $Dock8^{-/-}$ MEFs, and examined the effect of DOCK8 deficiency on subcellular localization of EPAS1 by staining them with anti-EPAS1 antibody. The specificity of this antibody was verified by lack of the reactivity to $Epas1^{-/-}$ MEFs (Supplementary Fig. 7). We found that, while EPAS1 translocated to the nucleus in both types of MEFs in response to hypoxia, nuclear localization of EPAS1 in $Dock8^{-/-}$ MEFs was markedly augmented even under the steady state (Fig. 6a). This effect of

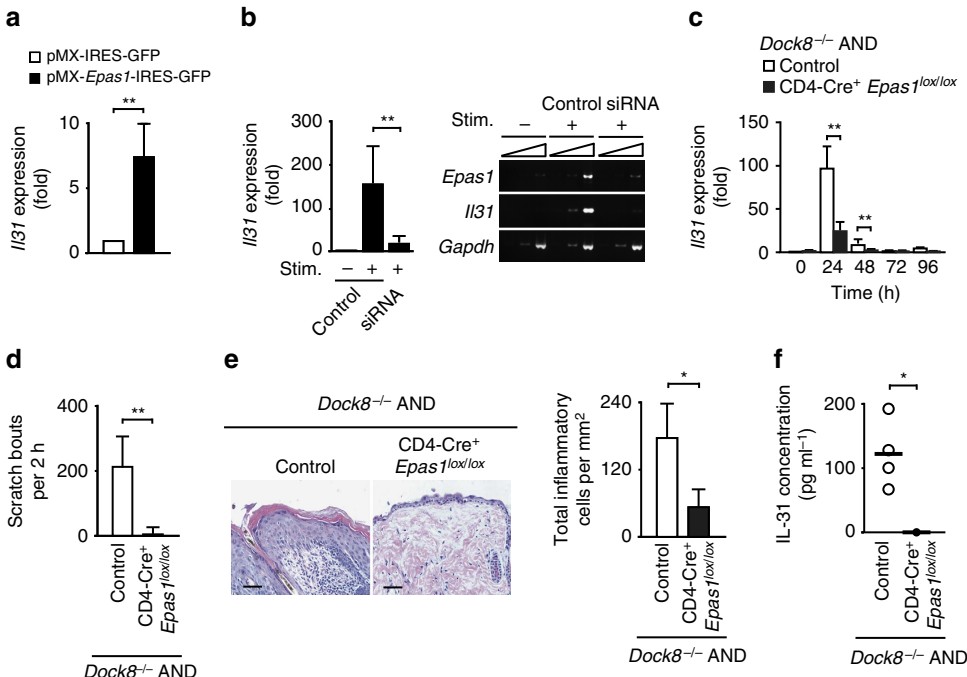

**Figure 4 | EPAS1 acts as a regulator for IL-31 induction during skin inflammation. (a)** Induction of *Il31* gene expression in WT CD4$^+$ T cells by overexpression of EPAS1. Expression (fold increase) is relative to that of the control vector (pMX-IRES-GFP). Data are expressed as mean ± s.d. of eight samples per group. **$P < 0.01$ (two-tailed Mann–Whitney test). **(b)** Effect of *Epas1* knock down on *Il31* gene expression in CD4$^+$ T cells from *Dock8*$^{-/-}$ AND Tg mice. Expression (fold increase) is relative to that of the unstimulated samples. Data are expressed as mean ± s.d. of 6 samples per group. **$P < 0.01$ (two-tailed Student's t-test). Amplification increased by 2 cycles, from the left to the right, starting at 34 cycles for *Epas1* and *Il31*, or at 29 cycles for *Gapdh*. **(c)** Effect of genetic inactivation of *Epas1* on *Il31* gene expression in CD4$^+$ T cells from *Dock8*$^{-/-}$ AND Tg mice after primary stimulation with MCC peptide. Expression (fold increase) is relative to that of the unstimulated control samples. Data are expressed as mean ± s.d. of 9 samples per group. **$P < 0.01$ (two-tailed Student's t-test). **(d)** Comparison of scratching bouts between CD4-Cre$^+$ *Epas1*$^{lox/lox}$*Dock8*$^{-/-}$ AND Tg mice and control littermates at 14 weeks old. Data are expressed as mean ± s.d. of five mice per group. **$P < 0.01$ (two-tailed Student's t-test). **(e)** Haematoxylin and eosin staining of the skin from 14-week-old littermates. Scale bars, 50 μm. The number of inflammatory cells per mm$^2$ was compared between CD4-Cre$^+$ *Epas1*$^{lox/lox}$*Dock8*$^{-/-}$ AND Tg mice and control littermates. In each experiment, 3 fields were examined per mouse. Data are expressed as mean ± s.d. of 4 mice group. *$P < 0.05$ (two-tailed Mann–Whitney test). **(f)** Serum concentrations of IL-31 in CD4-Cre$^+$ *Epas1*$^{lox/lox}$*Dock8*$^{-/-}$ AND Tg mice ($n = 4$) and control littermates ($n = 4$) at 14 weeks old. The lines indicate the mean values. *$P < 0.05$ (two-tailed Mann–Whitney test). In **(c–f)**, CD4-Cre$^-$ *Epas1*$^{lox/lox}$*Dock8*$^{-/-}$ AND Tg mice or CD4-Cre$^+$ *Epas1*$^{lox/w}$*Dock8*$^{-/-}$ AND Tg mice were used as controls.

DOCK8 deficiency was cancelled when WT DOCK8 was stably expressed in *Dock8*$^{-/-}$ MEFs (Fig. 6b,c). Similar results were obtained by expressing the DOCK8 mutant (ΔDHR2) lacking DOCK homology region (DHR)-2 domain critical for Cdc42 activation[6] (Fig. 6b,c). However, the expression of DOCK8 mutant (ΔN) lacking the N-terminal 527 amino acid residues in *Dock8*$^{-/-}$ MEFs failed to suppress nuclear accumulation of EPAS1 (Fig. 6b,c), indicating that the N-terminal region of DOCK8 is important for controlling subcellular localization of EPAS1.

MST1 is a serine/threonine kinase that has been implicated in T cell adhesion, migration, proliferation and apoptosis[33]. During the course of screening for DOCK8-binding proteins, we found that DOCK8 bound to MST1 through the N-terminal region when overexpressed in human embryonic kidney 293T (HEK-293T) cells (Fig. 6d, left). Similar association was observed with the activated CD4$^+$ T cells (Fig. 6e). When DOCK8, MST1 and EPAS1 were co-expressed in HEK-293T cells, EPAS1 and MST1 were co-immunoprecipitated, irrespective of the presence of DOCK8 (Fig. 6d, right). This raised the possibility that DOCK8 might regulate subcellular localization of EPAS1 through the association with MST1. Indeed, we found that nuclear translocation of EPAS1 was significantly augmented by knocking down *Mst1* gene expression (Fig. 6f). More importantly, knock down of *Mst1* markedly induced *Il31* gene

expression in CD4$^+$ T cells from *Dock8*$^{+/-}$ AND Tg mice (Fig. 6g). These results indicate that DOCK8-MST1 axis negatively regulates IL-31 induction by inhibiting nuclear translocation of EPAS1.

**Role of DOCK8 and EPAS1 in human CD4$^+$ T cells.** Finally, we examined the role of DOCK8 and EPAS1 in human CD4$^+$ T cells obtained from healthy controls and AD patients. In CD4$^+$ T cells from healthy controls, knock down of *DOCK8* gene markedly increased *IL31* gene expression (Fig. 7a), indicating that DOCK8 also acts as a negative regulator for IL-31 induction in human CD4$^+$ T cells. Consistent with this, a DOCK8-deficient patient[34] exhibited an elevated serum IL-31 level, as seen in conventional AD patients (Fig. 7b). Although DOCK8 expression was comparable between healthy controls and AD patients (Supplementary Fig. 8), TCR stimulation-induced expression of *IL31*, but not *IL2*, was much higher in CD4$^+$ T cells from AD patients than those from healthy controls (Fig. 7c). Importantly, induction of *IL31* gene expression was also cancelled when *EPAS1* gene expression was knocked down in CD4$^+$ T cells from AD patients (Fig. 7d). These results indicate that EPAS1-dependent pathway also operates for IL-31 induction in CD4$^+$ T cells from AD patients. To further examine the role of EPAS1 in

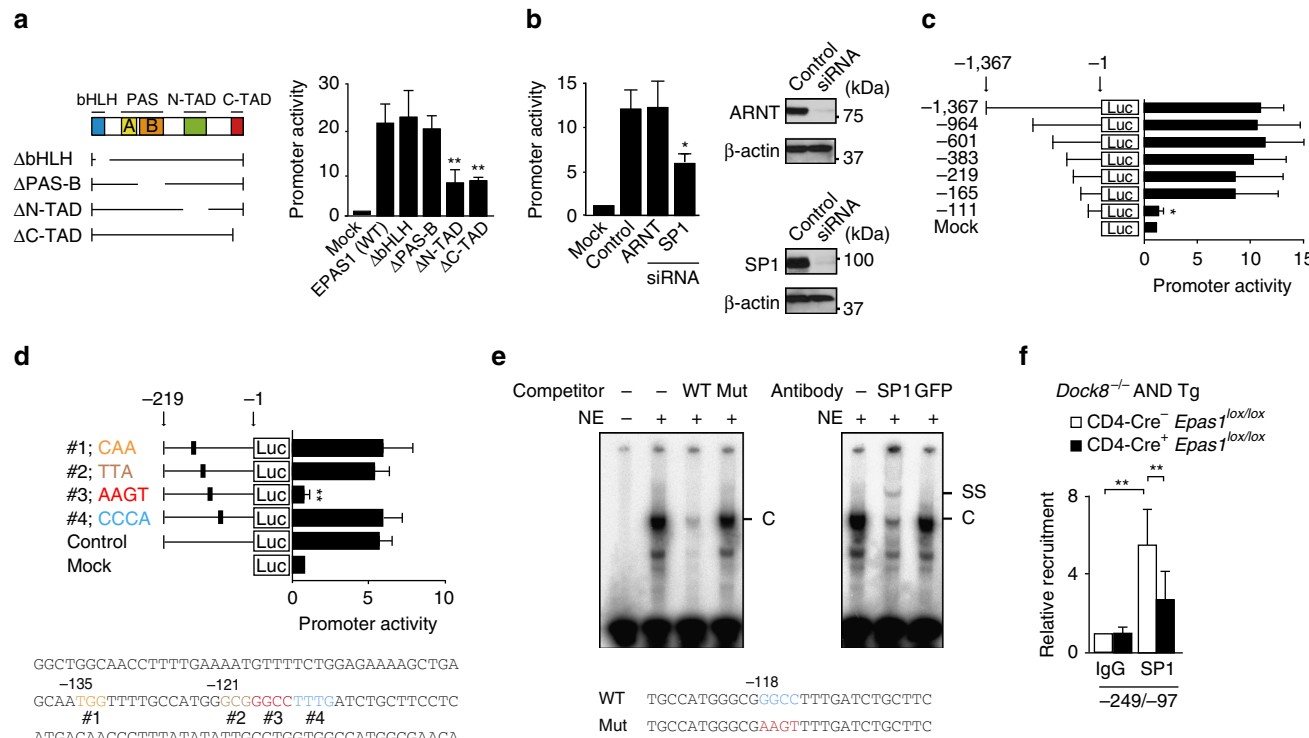

**Figure 5 | EPAS1 induces *Il31* promoter activation in collaboration with SP1.** (**a**) EPAS1-dependent *Il31* promoter activation. Schematic representation of EPAS1 mutants used in this study is shown. Data are expressed as mean ± s.d. of 5 independent experiments. **$P < 0.01$ (one-way analysis of variance (ANOVA) followed by *post hoc* Bonferroni test). (**b**) Requirement for SP1, but not ARNT, in EPAS1-mediated *Il31* promoter activation. Data are expressed as mean ± s.d. of three independent experiments. *$P < 0.05$ (one-way ANOVA followed by *post hoc* Bonferroni test). Knock down efficacy was checked by Western blot analyses. See Supplementary Fig. 11a for uncropped scans of the Westen blot. (**c,d**) Identification of the critical region of *Il31* promoter for EPAS1-mediated transactivation by deletion (**c**) and site-directed mutagenesis (**d**) studies. Data are expressed as mean ± s.d. of three independent experiments. *$P < 0.05$; **$P < 0.01$ (one-way ANOVA followed by *post hoc* Bonferroni test). (**e**) EMSA showing SP1 binding to the specific *Il31* promoter sequence. The 'C' or 'SS' indicates the band corresponding to SP1-DNA complex or super-shift, respectively. NE, nuclear extracts. Data are representative of 4 independent experiments. (**f**) ChIP assays showing recruitment of SP1 to *Il31* promoter in *Dock8*[−/−] AND Tg CD4[+] T cells and its reduction in the absence of EPAS1. Data (mean ± s.d., $n = 7$) are expressed as the relative recruitment after normalization of the level of control samples (CD4-Cre[−] *Epas1*[lox/lox] samples precipitated with rabbit IgG) to an arbitrary value of 1. **$P < 0.01$ (two-tailed Student's *t*-test). In **a–d**, promoter activity is expressed as the relative index after normalization of the luciferase activity of pGL4.10-*Il31* alone (Mock) to an arbitrary value of 1.

IL-31 induction, we treated CD4[+] T cells from AD patients with two inhibitors, FM19G11 and HIFVII. FM19G11 is not a direct inhibitor of EPAS1, but it is known to suppress *EPAS1* gene expression by acting on undefined target[35]. Indeed, the expression level of EPAS1 was markedly reduced when human CD4[+] T cells were treated with FM19G11 (Fig. 7e). In agreement with genetic data, we found that FM19G11 treatment ablated TCR stimulation-induced expression of *IL31*, but not *IL2*, in CD4[+] T cells from AD patients (Fig. 7f). However, such inhibitory effect was not observed with HIFVII (Fig. 7f), which binds to PAS-B domain of EPAS1 and inhibits its association with ARNT[26,36] (Supplementary Fig. 9). Thus, in CD4[+] T cells from AD patients, EPAS1 also regulates IL-31 induction independently of ARNT binding.

## Discussion

IL-31 is a T-cell derived cytokine implicated in pruritus in AD, yet the mechanism controlling IL-31 production remains unknown. Here we have identified EPAS1 as a key transcription factor that regulates IL-31 induction in CD4[+] T cells in both mice and humans. Although EPAS1 is known to form a complex with ARNT and control various biological functions such as vascular remodelling, pulmonary development and skeletal growth[37–42], EPAS1 mediated *Il31* promoter activation

independently of ARNT, but in collaboration with SP1. A similar ARNT-independent and SP1-dependent promoter activation by EPAS1 was reported for a few genes such as copper-transporting ATPase (*Atp7a*) gene and blood coagulation factor VII (*fVII*) gene[43,44]. The precise mechanism by which SP1 potentiates EPAS1-mediated *Il31* promoter activation remains to be determined. However, we found that SP1 recruitment to *Il31* promoter was significantly reduced in EPAS1-deficient CD4[+] T cells. Therefore, it seems likely that EPAS1 recruits SP1 and other co-factors to form 'EPAS1 enhanceosomes' and alters the mode of transcription from a basal to an inducible state, as previously suggested[45].

Homozygous and compound heterozygous mutations in *DOCK8* cause a combined immunodeficiency characterized by recurrent viral infections, early onset malignancy and AD[1–5]. However, the molecular link between DOCK8 deficiency and atopic skin inflammation is poorly understood. In this study, we have shown that *Dock8*[−/−] CD4[+] T cells produce large amounts of IL-31 in a manner dependent on EPAS1. Consistent with this, a DOCK8-deficient patient[34] exhibited increased serum IL-31. Mechanistically, we found that nuclear translocation of EPAS1 was significantly augmented in the absence of DOCK8. Although DOCK8 is a Cdc42-specific GEF[6], rescue experiments revealed that the GEF activity of DOCK8 was not required in this process. Instead, DOCK8

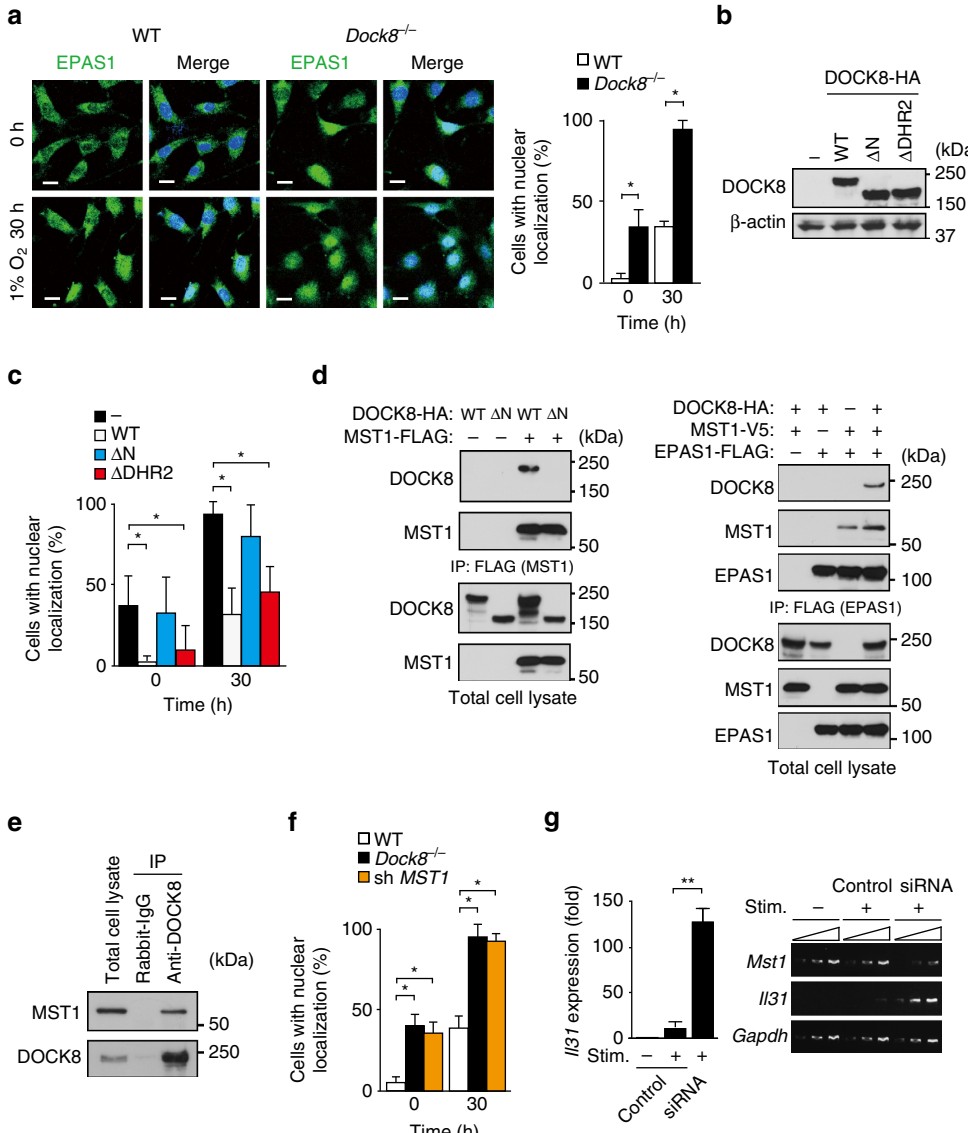

**Figure 6 | DOCK8 regulates nuclear translocation of EPAS1 via MST1 interaction.** (**a**) Immunofluorescence analyses for subcellular localization of EPAS1 in MEFs under steady and hypoxia conditions. DAPI was used to stain nuclei. Scale bars, 20 μm. The proportion of cells with nuclear localization of EPAS1 was compared between WT and $Dock8^{-/-}$ MEFs. In each experiment, 30 cells were counted per group. Data are expressed as mean ± s.d. of 4 independent experiments. *$P<0.05$ (two-tailed Mann–Whitney test). (**b,c**) Comparison of nuclear localization of EPAS1 between $Dock8^{-/-}$ MEFs and those stably expressing HA-tagged WT DOCK8 or its mutants (ΔN and ΔDHR2). Western blot analysis was performed with anti-HA and anti-βactin antibodies. In each experiment, 30 cells were counted per group. Data are expressed as mean ± s.d. of four independent experiments. *$P<0.05$ (two-tailed Mann–Whitney test). (**d**) Formation of DOCK8-MST1-EPAS1 molecular complex in HEK-293T cells. 24 h after transfection of the expression plasmids in HEK-293T cells, cell extracts were subjected to immunoprecipitation and immunoblotting. Data are representative of three separate experiments. (**e**) Association between DOCK8 and MST1 in activated CD4$^+$ T cells. The cell extracts of activated CD4$^+$ T cells were immunoprecipitated with anti-DOCK8 antibody or control rabbit IgG. Immunoblotting was carried out with anti-MST1 and anti-DOCK8 antibodies. Data are representative of three separate experiments. (**f**) Effect of $Mst1$ knock down on nuclear localization of EPAS1 in WT MEFs. In each experiment, 30 cells were counted per group. Data are expressed as mean ± s.d. of four independent experiments. *$P<0.05$ (two-tailed Mann–Whitney test). (**g**) Effect of $Mst1$ knock down on $Il31$ gene expression in CD4$^+$ T cells from $Dock8^{+/-}$ AND Tg mice. Expression (fold increase) is relative to that of the unstimulated samples. Data are expressed as mean ± s.d. of 6 samples per group. **$P<0.01$ (two-tailed Student's $t$-test). Amplification increased by 2 cycles, from the left to the right, starting at 34 cycles for $Mst1$ and $Il31$ or at 29 cycles for $Gapdh$. See Supplementary Fig. 11b–d for uncropped scans of the westen blot.

acted as an adaptor and negatively regulated nuclear translocation of EPAS1 through the interaction with MST1. Interestingly, it has been reported that 7 of 9 patients with $MST1$ mutations had eczema or AD-like skin disease[46–48], raising the possibility that MST1 is also involved in IL-31 induction. Indeed, we found that $Il31$ gene expression markedly increased in CD4$^+$ T cells when $Mst1$ gene was knocked down. Our results thus

define a previously unknown signalling cascade that links DOCK8 deficiency to IL-31 induction.

Although CD4$^+$ T cells from both $Dock8^{-/-}$ AND Tg and $Dock8^{-/-}$ OTII Tg mice produced IL-31 on stimulation with cognate antigens, only $Dock8^{-/-}$ AND Tg mice spontaneously developed atopic skin disease. It is clear that CD4$^+$ T cells play a major role in the disease development,

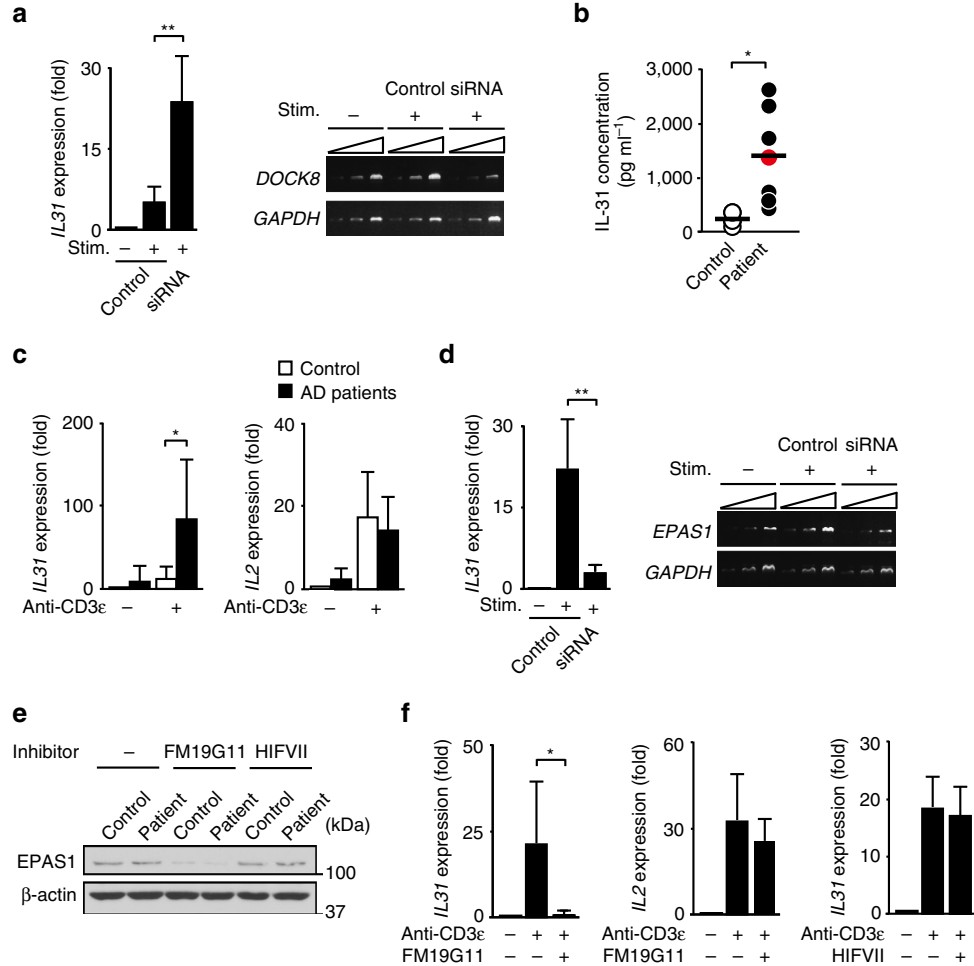

**Figure 7 | Role of DOCK8 and EPAS1 in IL-31 induction in human CD4$^+$ T cells.** (**a**) Effect of *DOCK8* knock down on TCR stimulation-induced *IL31* gene expression in CD4$^+$ T cells from healthy controls. Expression (fold increase) is relative to that of the unstimulated samples. Data are expressed as mean ± s.d. of eight samples. **$P < 0.01$ (two-tailed Student's *t*-test). Amplification increased by 2 cycles, from the left to the right, starting at 34 cycles for *DOCK8* or at 29 cycles for *GAPDH*. (**b**) Comparison of serum IL-31 concentrations between healthy controls ($n = 6$) and DOCK8-deficient patient ($n = 1$; red circle) or AD patients ($n = 6$). The lines indicate the mean values. *$P < 0.05$ (two-tailed Student's *t*-test for comparison between AD patients and healthy controls). (**c**) TCR stimulation-induced cytokine gene expression in CD4$^+$ T cells from AD patients ($n = 6$) and healthy controls ($n = 6$). Expression (fold increase) is relative to that of unstimulated healthy controls. Data are expressed as mean ± s.d. *$P < 0.05$ (two-tailed Mann–Whitney test). (**d**) Effect of *EPAS1* knock down on *IL31* gene expression in CD4$^+$ T cells from AD patients. Expression (fold increase) is relative to that of the unstimulated samples. Data are expressed as mean ± s.d. of 6 samples. **$P < 0.01$ (two-tailed Mann–Whitney test). Amplification increased by 2 cycles, from the left to the right, starting at 34 cycles for *EPAS1* or at 29 cycles for *GAPDH*. (**e**) Effect of on FM19G11 and HIFVII on EPAS1 expression in CD4$^+$ T cells from AD patients and healthy controls. Data are representative of three separate experiments. See Supplementary Fig. 11e for uncropped scans of the Westen blot. (**f**) Effect of FM19G11 ($n = 6$) and HIFVII ($n = 3$) on TCR stimulation-induced cytokine gene expression in CD4$^+$ T cells from AD patients. Expression (fold increase) is relative to that of AD patient samples without stimulation. Data are expressed as mean ± s.d. *$P < 0.05$ (two-tailed Student's *t*-test).

because all the phenotypes disappeared in *Dock8*$^{-/-}$ AND Tg mice when EPAS1 expression was specifically deleted in CD4$^+$ T cells. The precise reason for this discrepancy is currently unknown. However, since CD4$^+$ T cells from *Dock8*$^{-/-}$ OTII Tg mice also induced itch *in vivo* on transfer into CAG-OVA mice, antigen availability may be a determinant of the disease induction. Unlike OTII TCR, AND TCR is a product of artificial αβ-chain combinations, and recent evidence indicates that AND TCR has much higher 'self-reactivity' to I-A$^b$ molecules than OTII TCR[49]. Therefore, it might be possible that CD4$^+$ T cells expressing AND TCR are continuously, albeit weakly, stimulated with self-peptide/I-A$^b$ complexes *in vivo*, and induce skin inflammation by producing IL-31 only when DOCK8 expression is lacking.

AD is a chronic inflammatory skin disease, and its prevalence is steadily increasing all over the world[50]. In this study, we have also shown that genetic or pharmacological inactivation of EPAS1 suppresses IL-31 induction in CD4$^+$ T cells from AD patients. As DOCK8 expression is unchanged between AD patients and healthy controls, it is likely that DOCK8-independent, but EPAS1-dependent mechanism operates for IL-31 induction in AD patients. How EPAS1 is activated in CD4$^+$ T cells from AD patients is currently unknown. However, recent evidence indicates that in human CD4$^+$ T cells, EPAS1 is a direct target of STAT6 and serves as a hub protein in IL-4-mediated transcription circuitries[27,28]. Therefore, it is highly conceivable that multiple genetic and environmental factors skewing Th2 differentiation could contribute to EPAS1 activation in AD patients. Whether particular cascades are involved in EPAS1 activation in AD patients would be an important issue that should be investigated in future studies.

In conclusion, we have demonstrated that EPAS1 plays a key role in IL-31 induction in mice and humans with atopic skin inflammation. Thus, EPAS1 may be a therapeutic target for controlling IL-31-associated itch.

## Methods

**Mice.** $Dock8^{-/-}$ mice have been described previously[6]. Mice heterozygous for the mutant allele ($Dock8^{+/-}$) were backcrossed onto a C57BL/6 background for more than nine generations, and $Dock8^{+/-}$ mice were crossed with AND TCR Tg mice or OTII TCR Tg mice to obtain $Dock8^{+/-}$ or $Dock8^{-/-}$ TCR Tg mice. $Osmr^{-/-}$ mice were obtained from RIKEN BioResource Center (RBRC02711). $Epas1^{lox/lox}$ mice were obtained from the Jackson Laboratory and crossed with CD4-Cre Tg mice to obtain CD4-Cre$^+$ $Epas1^{lox/lox}Dock8^{-/-}$ AND Tg mice. CAG-OVA mice were purchased from the Jackson Laboratory. Mice were maintained under specific-pathogen-free conditions in the animal facility of Kyushu University, and age- and sex-matched littermates were used as controls. Animal experiments were approved by the committee of Ethics of Animal Experiments, Kyushu University. Mice were selected randomly and assigned to experimental groups according to genotype. The investigators who performed the experiments were not blinded to mouse genotypes.

**Measurement of scratching behaviour.** Mice were put into an acrylic cage ($11 \times 14 \times 20$ cm) for at least 1 h for acclimation, and their behaviours were videotaped. Playback of the video was used for determination of the total number of scratching bouts per 2 h. When mice scratch, they stretch their hind paw toward the itchy spot, lean the head toward the hind paw, rapidly move the paw several times, and then lower it back to the floor. A series of these movements was counted as one bout of scratching, as previously described[51].

**Evaluation of skin disease.** The severity of dermatitis was grossly assessed by the modified SCORAD method, as described previously[31]. This scoring is based on the severity of (1) erythema/hemorrhage, (2) scaling/dryness, (3) oedema and (4) excoriation/erosion. The degree of each symptom was scored as 0 (absence), 1 (mild), 2 (moderate) and 3 (severe). The sum of the individual scores (minimum 0 and maximum 12) was taken as the dermatitis score.

**Histology and immunofluorescence.** Skin tissues were fixed in 4% (w/v) paraformaldehyde and embedded in paraffin blocks. Sections (3 μm thick) were stained with haematoxylin and eosin, and examined by light microscopy. For immunofluorescence analyses, tissues were embedded in OCT compound (Sakura Finetech) and frozen in liquid nitrogen. Cryostat sections (10 μm thick) were fixed in 4% paraformaldehyde for 30 min and blocked with 10% goat serum for 1 h at room temperature. Sections were then stained with phycoerythrin (PE)-conjugated anti-mouse CD3 (17A2, 4 μg ml$^{-1}$; Biolegend), fluorescein isothiocyanate (FITC)-conjugated anti-mouse CD45R/B220 (RA3-6B2, 5 μg ml$^{-1}$; BD Biosciences), FITC-conjugated anti-mouse CD8a (53–6.7, 5 μg ml$^{-1}$; BD Biosciences), and biotinylated anti-mouse CD4 (H129.19, 5 μg ml$^{-1}$; BD Biosciences) antibodies followed by incubation with Alexa Fluor 488-conjugated streptavidin (4 μg ml$^{-1}$; Thermo Fisher Scientific). For EPAS1 staining, MEFs ($3 \times 10^5$ cells per ml) were cultured on the poly-L-lysine coated glass-bottom dishes (Matsunami) in 1% $O_2$ for 30 h, fixed with 4% (w/v) paraformaldehyde for 30 min and permeabilized with 0.2% Triton X-100 for 30 min. After being blocked with 10% goat serum for 1 h at room temperature, cells were then stained with 4′,6-diamidino-2-phenylindole (DAPI; Dojindo Laboratories) and rabbit anti-EPAS1 antibody (#100–122, 10 μg ml$^{-1}$; NOVUS Biologicals), followed by incubation with Alexa Fluor 488-conjugated donkey anti-rabbit IgG Fab fragment (#711–547-003, 3 μg ml$^{-1}$; Jackson ImmunoResearch) antibody. All images were obtained with a laser scanning confocal microscope (Carl Zeiss).

**ELISA.** IL-31 concentrations in serum samples and cell culture supernatants were measured with ELISA kits (R&D Systems for human samples and USCN for mouse samples), according to the manufacture's instructions. To measure the concentrations of serum IgE and IgG2b, serum samples were serially diluted and placed in 96-well plates coated with goat anti-mouse IgE (#1110–01) or goat anti-mouse Ig (IgM + IgG + IgA, H + L; #1010-01) antibody (both 2.5 μg ml$^{-1}$; Southern Biotech). After 2 h incubation, wells were washed with phosphate-buffered saline (PBS) and incubated with alkaline phosphatase-conjugated rat anti-mouse IgE (#1130–04, 1:3,000 dilution; Southern Biotech) or goat anti-mouse IgG2b (#1090-04, 1:100,000 dilution; Southern Biotech) antibody.

**Flow cytometry.** The following antibodies and reagents were used. FITC-conjugated anti-mouse CD45R/B220 (RA3-6B2, 5 μg ml$^{-1}$), FITC-conjugated anti-mouse CD4 (RM4–5, 5 μg ml$^{-1}$), biotinylated anti-mouse CD4 (RM4–5, 2.5 μg ml$^{-1}$), PE-conjugated anti-mouse CD8a (53-6.7, 1 μg ml$^{-1}$), PE-conjugated anti-mouse CD44 (IM7, 1 μg ml$^{-1}$), FITC-conjugated anti-mouse CD62L (MEL-14, 5 μg ml$^{-1}$), FITC-conjugated anti-mouse Vα11

(RR8-1, 5 μg ml$^{-1}$), FITC-conjugated anti-mouse Vα2 (B20.1, 5 μg ml$^{-1}$), biotinylated anti-mouse Vβ3 (KJ25, 2.5 μg ml$^{-1}$), biotinylated anti-mouse Vβ5 (MR9-4, 2.5 μg ml$^{-1}$), and allophycocyanin (APC)-conjugated streptavidin (0.1 μg ml$^{-1}$) or PerCP-5.5cyanine-conjugated streptavidin (0.5 μg ml$^{-1}$) were from BD Biosciences. Biotinylated anti-mouse CD90.2/Thy1.2 (30-H12, 0.25 μg ml$^{-1}$) was purchased from eBioscience. Before staining with the antibodies, cells were incubated for 10 min on ice with anti-Fcγ III/II receptor (2.4G2, 0.5 μg ml$^{-1}$; BD Biosciences) antibody to block Fc receptors. Flow cytometoric analyses were done on FACS Calibur (BD Biosciences).

**Activation and transfer of mouse CD4$^+$ T cells.** Mouse CD4$^+$ T cells were isolated from the spleen and peripheral lymph nodes (PLNs) by magnetic sorting with Dynabeads mouse CD4 followed by treatment with DETACHaBEAD mouse CD4 (Life Technologies), and suspended in RPMI 1640 medium (Wako) containing 10% heat-inactivated fetal calf serum (FCS; Nichirei Bioscience), 50 μM 2-mercaptoethanol (Nacalai tesque), 2 mM L-glutamine (Life Technologies), 100 U ml$^{-1}$ penicillin (Life Technologies), 100 μg ml$^{-1}$ streptomycin (Life Technologies), 1 mM sodium pyruvate (Life Technologies) and MEM non-essential amino acids (Life Technologies). For primary stimulation, CD4$^+$ T cells ($3 \times 10^5$ cells per well) were cultured in a 24-well plate with T cell-depleted, irradiated spleen cells ($5 \times 10^6$ cells per well) in the presence of the following peptides: MCC88–103 (ANERADLIAYLKQATK, 3 μg ml$^{-1}$) or OVA323–339 (ISQAVHAAHAEINEAGR, 1 μg ml$^{-1}$). In some experiments, activated CD4$^+$ T cells ($5.7 \times 10^6$ per mouse) from $Dock8^{+/-}$ and $Dock8^{-/-}$ OTII Tg mice were intravenously injected into CAG-OVA mice 5 h before assays. For secondary stimulation, CD4$^+$ T cells recovered from the culture were re-stimulated with plate-bound anti-CD3ε antibody (145-2C11, 1 μg ml$^{-1}$; eBioscience) and anti-CD28 antibody (37.51, 1 μg ml$^{-1}$; BD Biosciences) for specified times. T cell proliferation assays were done by cultivating CD4$^+$ T cells ($5 \times 10^4$ per well) with T cell-depleted, irradiated spleen cells ($1 \times 10^6$ cells per well) in the presence or absence of various concentrations of the relevant peptide for 66 h. [$^3$H]-thymidine (0.037 MBq) was added during the final 18 h of the culture, and the incorporated radioactivity was measured with a liquid scintillation counter.

**Preparation of MEFs.** Primary MEFs were generated from WT, $Dock8^{-/-}$ mice and $Epas1^{lox/lox}$ mice at E13.5, and $Epas1$ gene expression was deleted in $Epas1^{lox/lox}$ MEFs by expressing pIC-Cre encoding Cre recombinase (provided by K. Takeda, Osaka University, Osaka, Japan). These MEFs were immortalized by transfection with a plasmid pCX4bsr-SV40ER (provided by T. Akagi, KAN Research Institute, Kobe, Japan) before use. Immortalized MEFs were cultured in DMEM medium (Wako Pure Chemical Industries) supplemented with 10% heat-inactivated FCS (Nichirei Bioscience), 100 U ml$^{-1}$ penicillin (Life Technologies) and 100 μg ml$^{-1}$ streptomycin (Life Technologies).

**Preparation and activation of human CD4$^+$ T cells.** Human peripheral blood mononuclear cells (PBMCs) were separated by Percoll (GE Healthcare) gradient centrifugation, and CD4$^+$ T cells were isolated from PBMCs by magnetic sorting with Dynabeads human CD4 followed by treatment with DETACHaBEAD human CD4 (Life Technologies). After being suspended in complete RPMI 1640 medium, CD4$^+$ T cells ($3 \times 10^5$ cells per well) were stimulated in a 24-well plate coated with anti-human CD3ε antibody (Hit3a, 10 μg ml$^{-1}$; Tonbo Biosciences) for 3 or 6 h. In some experiments, CD4$^+$ T cells were treated with EPAS1 inhibitors, FM 19G11 or HIFVII at 30 μM (both from Calbiochem). Human peripheral blood samples were obtained from a DOCK8-deficient patient[34], AD patients and healthy volunteers in compliance with Institutional Review Board protocols. AD was diagnosed according to the criteria provided by the Japanese Dermatological Association[52]. The details for the AD patients and healthy controls are shown in Supplementary Table 2. The experiments were approved by the Ethics committee of Kyushu University Hospital and Fujita Health University, and a written informed consent was obtained from all patients.

**Reverse transcription–PCR.** Total RNA was isolated using ISOGEN (Nippon Gene). After treatment with RNase-free DNase I (Life Technologies), RNA samples were reverse-transcribed with oligo(dT) primers (Life Technologies) and SuperScript III reverse transcriptase (Life Technologies) for amplification by PCR. The primers used for conventional reverse transcription–PCR and real-time PCR are listed in Supplementary Table 3. Real-time PCR was performed on an ABI PRISM 7,000 Sequence Detection System using the SYBR Green PCR Master Mix (both from Applied Biosystems). The expressions of human and mouse target genes were normalized to expression of GAPDH and Hprt, respectively. Sequence-detection software supplied with the instrument was used for analyses.

**Microarray analysis.** Total RNA was isolated using ISOGEN (Nippon Gene), and cRNA was amplified and labelled using a Low Input Quick Amp Labeling Kit (Agilent Technologies). The cRNA was then hybridized to a 44 K 60-mer oligomicroarray (Whole Mouse Genome oligo DNA Microarray Kit Ver 2.0; Agilent Technologies). The hybridized microarray slides were scanned using an

Agilent scanner. The relative hybridization intensities and background hybridization values were calculated using Feature Extraction Software version 9.5.1.1 (Agilent Technologies). Raw signal intensities and flags for each probe were calculated from the hybridization intensities and spot information, according to the procedures recommended by Agilent Technologies. To identify up- or down-regulated genes in experimental samples, we calculated $Z$-scores and ratios from the normalized signal intensities of each probe (upregulated genes, $Z$-score $> 2.0$ and ratio $> 1.5$-fold; down-regulated genes, $Z$-score $< -2.0$ and ratio $< 0.66$-fold).

**Knock down of target genes by siRNAs.** To knock down *Epas1* or *Mst1* gene expression in *Dock8*$^{-/-}$ or *Dock8*$^{+/-}$ AND CD4$^+$ T cells, the Accell siRNA SMART pool, E-040635-00-0005 or E-059385-00-0005 (both from Dharmacon) was used, respectively. Transfection was performed according to the manufacturer's instructions using the irrelevant oligonucleotide (Accell Red Non-targeting siRNA; D-001960-01-05, Dharmacon) as a control. Briefly, *Dock8*$^{-/-}$ or *Dock8*$^{+/-}$ AND CD4$^+$ T cells ($3 \times 10^5$ cells per well) were cultured with T cell-depleted, irradiated spleen cells ($5 \times 10^6$ cells per well) and MCC88–103 peptide ($3 \mu g ml^{-1}$) in Accell siRNA Delivery Media (Dharmacon) supplemented with 2.5% FCS. The siRNA or the irrelevant oligonucleotide was added to the culture at the final concentration of $1 \mu M$. After 4 days of the culture, viable CD4$^+$ T cells were recovered and re-stimulated with plate-bound anti-CD3ε (145-2C11, $1 \mu g ml^{-1}$) and anti-CD28 (37.51, $1 \mu g ml^{-1}$) antibodies for 3 h.

Knock down of *EPAS1* or *DOCK8* gene expression in human CD4$^+$ T cells was performed with the Accell siRNA SMART pool, E-004814-00-0005 or E-026106-00-0005 (both from Dharmacon) with irrelevant oligonucleotide (Accell Red Non-targeting siRNA; D-001960-01-05, Dharmacon) as a control. For this purpose, human PBMCs ($1 \times 10^6$ cells per well) were cultured with staphylococcal enterotoxin B ($0.1 \mu g ml^{-1}$; Sigma-Aldrich) in Accell siRNA Delivery Media (Dharmacon) supplemented with 2.5% FCS. The siRNA or control oligonucleotide was added to the culture at the final concentration of $1 \mu M$. After 4 days (for *EPAS1* knock-down) or 2 days (*DOCK8* knock-down) of the culture, viable CD4$^+$ T cells were recovered by magnetic sorting and re-stimulated with plate-bound anti-CD3ε (Hit3a, $10 \mu g ml^{-1}$) antibody for 3 h. In these experiments, knock down efficacy was checked by reverse transcription–PCR.

To knock down *Sp1* and *Arnt* gene expression in MEFs, the siRNAs (On-Target plus SMART pool, L-040633-02-0005 and L-040639-01-0005) were obtained from Dharmacon, and the irrelevant oligonucleotide (BLOCK-iT Alexa Fluor Red; Life Technologies) was used as a negative control. For transfection, $400 \mu l$ of the siRNA solution (150 nM) in serum-free DMEM medium (Wako) was mixed with $5 \mu l$ DharmaFect transfection reagent for 20 min at room temperature. This mixture ($400 \mu l$) was added drop-wise to MEFs ($3 \times 10^5$ cells per well) suspended in 1.6 ml DMEM medium containing 10% FCS. Then, cells were incubated for 24 h at 37 °C before transient transfection for luciferase reporter assays. Knock down efficacy was checked by Western blot analyses.

**Plasmids and transfection.** To generate the *Il31* reporter plasmid (pGL4.10-*Il31*), the promoter region of the mouse *Il31* gene from position $-1,367$ to $-1$ was amplified by PCR and subcloned into pGL4.10[luc2] vector (Promega). Deletions and mutations in the *Il31* promoter region were created by PCR. These plasmids were transfected into MEFs with Lipofectamine 2,000 reagent (Life Technologies) for luciferase reporter assays. The pcDNA vector (Invitrogen) was used to create expression vectors encoding FLAG-tagged human and mouse EPAS1 (pcDNA-EPAS1) or its deletion mutants, HA-tagged mouse DOCK8 (pcDNA-DOCK8) or its mutant, FLAG- or V5-tagged mouse MST1 (pcDNA-MST1), V5-tagged mouse SP1 (pcDNA-SP1) and HA-tagged human and mouse ARANT (pcDNA-ARNT). These expression constructs were transfected into HEK-293T cells with polyethyleminine for immunoprecipitation. The pBJ vector encoding neomycin (pBJ-neo) was used to express HA-tagged WT mouse DOCK8 and its mutants, ΔN lacking the N-terminal 527 amino acid residues and ΔDHR2 lacking amino acid residues from 1,535 to 2,100. The retroviral vector pMX was used to generate the plasmid encoding pMX-*Epas1*-IRES-GFP. This plasmid DNA was transfected into Platinum-E packaging cells with FuGENE 6 transfection reagent (Promega). The cell culture supernatants were harvested 48 h after transfection, supplemented with polybrene ($5 \mu g ml^{-1}$) and IL-2 ($5 ng ml^{-1}$), and were used to infect CD4$^+$ T cells stimulated with plate-bound anti-CD3ε and anti-CD28 antibodies. After centrifugation at $700 g$ for 1 h, plates were incubated for 8 h at 32 °C and for 16 h at 37 °C. Two additional retroviral infections were performed at daily intervals, and the GFP-positive CD4$^+$ T cells were sorted by FACSAria (BD Biosciences) for RNA extraction 30 h after the third transfection. To generate the retroviral vector pSUPER retro-puro sh *Mst1*, the following oligonucleotides corresponding to specific regions of *Mst1* gene were ligated to the pSUPER retro-puro vector (OligoEngine) at the *Bgl* II and *Hind* III sites: 5′-GATCTCACCGCCAGATTGTTGCAATCAAGCCGAAGCTTGATTGCAA-CAATC TGGCTTTTA-3′ and 5′-AGCTTAAAAGCCAGATTGTTGCA ATCAAGCTTCGGCTT GATTGCAACAATCTGGCGGTGA-3′. MEFs were retrovirally transduced with pSUPER retro-puro sh *Mst1* as described above.

**Luciferase reporter assays.** MEFs were co-transfected with pRL-SV40-*Renilla* luciferase plasmid ($0.1 \mu g$; Promega) and pGL4.10-*Il31* ($2 \mu g$) in the presence of pcDNA-*Epas1* or its mutants ($2 \mu g$). For transfection, these plasmid DNAs were mixed with Lipofectamine 2,000 transfection reagent ($5 \mu l$) in the Opti-MEM medium ($500 \mu l$; Life Technologies) for 20 min at room temperature, and the mixture was added drop-wise to MEFs ($3 \times 10^5$ cells per well) cultured in 1.5 ml of DMEM medium containing 1% FCS. 6 h after transfection, cells were suspended in DMEM containing 10% FCS and incubated for additional 24 h. In some experiments, pGL4.10-*Il31*-derived mutants were used. The total amount of plasmid DNA was equalized by the control vector. Luciferase activity was measured with a Dual-Luciferase Reporter Assay System according to the manufacture's protocols (Promega).

**Immunoblotting and immunoprecipitation.** Cells were lysed on ice in 20 mM Tris-HCl buffer (pH 7.5) containing 1% Triton X-100, 150 mM NaCl, 1 mM EDTA, 1 mM EGTA, 2.5 mM sodium pyrophosphate, 1 mM β-glycerophosphate, 1 mM $Na_3VO_4$, and complete protease inhibitors (Roche). After centrifugation, the supernatants were mixed with an equal volume of $2 \times$ sample buffer (125 mM Tris-HCl, 0.01% bromophenol blue, 4% SDS, 20% glycerol and $200 \mu M$ dithiothreitol). Samples were boiled for 5 min and analysed by immunoblotting. The following antibodies were used: rabbit anti-EPAS1 (#100–122, 1:1,000 dilution; Novus Biologicals), rabbit anti-SP1 (#07–645, 1:1,000 dilution; Millipore), rabbit anti-ARNT (#5537, 1:1,000 dilution; Cell Signaling), rabbit anti-MST1 (#3682, 1:1,000 dilution; Cell Signaling), goat anti-βactin (I-19, 1:1,000 dilution; Santa Cruz), rat anti-HA (3F10, 1:2,000 dilution; Roche), anti-FLAG (#PM020-7, 1:4,000 dilution; MBL) and anti-V5 (R961-25, 1:2,000 dilution; Life Technologies). Polyclonal antibody against DOCK8 was produced by immunizing rabbits with KLH-coupled synthetic peptide corresponding to the C-terminal sequence of human and mouse DOCK8 (RDSFHRSSFRKCETQLSQGS, $3.8 \mu g ml^{-1}$). To examine association between DOCK8 and MST1 in activated CD4$^+$ T cells, cell extracts were immunoprecipitated with anti-DOCK8 antibody or control rabbit IgG (#011-000-002, Jackson ImmunoReseach). The bound proteins were analysed by SDS-PAGE, and blots were probed with rabbit anti-MST1 antibody (#3682, 1:1,000 dilution; Cell Signaling) followed by mouse anti-rabbit IgG light-chain specific antibody (L57A3, 1:1,000 dilution; Cell Signaling) and horseradish peroxidase (HRP)-conjugated goat anti-mouse IgG (#2005, 1:2000 dilution; Santa Cruz). In some experiments, HEK-293T cells transfected with the specified plasmid DNAs were subjected to immunoprecipitation using the relevant antibodies.

**EMSA.** Nuclear extracts were prepared from MEFs using the standard method. Briefly, cells were rinsed with PBS, resuspended in buffer A (10 mM HEPES-K$^+$, pH 7.9, 10 mM KCl, 0.1 mM EDTA, 0.1 mM EGTA, 0.5 mM DTT, 1 mM PMSF) and incubated on ice for 15 min. Cells were then lysed by adding NP-40 to a final concentration of 0.67% and immediately vortexing for 10 s. The lysate was centrifuged at $20,000 g$ for 30 s at 4 °C to pellet nuclei. Nuclei were resuspended in buffer B (20 mM HEPES-K$^+$, pH 7.9, 400 mM NaCl, 1 mM EDTA, 1 mM EGTA, 1 mM DTT, 1 mM PMSF), incubated for 15 min on ice with intermittent agitation and centrifuged at $20,000 g$ for 5 min at 4 °C to obtain nuclear extracts. A DNA probe corresponding to the mouse *Il31* promoter region containing the consensus SP1-binding sequence (WT; 5′- TGCCATGGGCGGGCCTTTGATCTGCTTC- 3′) was labelled with γ-$^{32}$P-ATP (Perkin Elmer) using T4 polynucleotide kinase (Promega) and purified using illustra$^{TM}$ MicroSpin$^{TM}$ G-25 columns (GE Healthcare). For protein-DNA binding, $4 \mu g$ nuclear extracts were incubated in $9 \mu l$ of binding buffer (20 mM HEPES-K$^+$, pH 7.9, 50 mM KCl, 3 mM $MgCl_2$, 10% glycerol, 1 mM DTT) supplemented with $2 \mu g$ poly(dI-dC) (Sigma-Aldrich) with or without unlabelled competitor DNA (WT or its mutant; 5′- TGCCATGGGC-GAAGTTTTG ATCTGCTTC-3′; 1.75 pmol) for 2.5 h on ice, before addition of $1 \mu l$ of $^{32}$P-labelled probe (0.035 pmol) and incubation at room temperature for 20 min. Protein-DNA complexes were separated on a 6% native polyacrylamide/ $0.5 \times$ TBE gel at 4 °C for 2 h, dried onto a filter paper at 80 °C for 2 h under vacuum and analysed with the BAS2,000 BIO Imaging Analyzer (Fuji Photo Film). For super-shift assays, $3 \mu g$ anti-SP1 antibody (#07–645; Millipore) or $6 \mu g$ anti-GFP antibody (A11122; Invitrogen) as a control were added to nuclear extracts and incubated on ice for 2.5 h before addition of radiolabelled probes.

**ChIP assay.** CD4$^+$ T cells were prepared from CD4-Cre$^+$ or CD4-Cre$^-$ *Epas1*$^{lox/lox}$*Dock8*$^{-/-}$ AND Tg mice and were stimulated with MCC88–103. After re-stimulation with plate-bound anti-CD3ε and anti-CD28 antibodies, cells were cross-linked with 1% formaldehyde for 10 min at room temperature and then neutralized with glycine for 5 min at room temperature. Cells were washed twice with PBS containing protease inhibitors, and then nuclei were isolated using the Magna ChIP$^{TM}$ HiSens kit (Millipore) according to the manufacturer's protocol. Isolated nuclei were resuspended in sonication buffer and sonicated using the Branson sonifier (Cell disruptor 200) with 16 sets of 10 pulses using the power set at 5.5 on ice for shearing chromatin DNA. After centrifugation at $10,000 g$ for 10 min at 4 °C, sheared chromatin DNA was obtained and treated with RNase and proteinase K for quantifying DNA content. For immunoprecipitation, $3.5–5 \mu g$ chromatin DNA was mixed with magnetic Protein A/G beads preincubated with $2 \mu g$ anti-SP1 antibody (#07–645; Millipore) or $1 \mu g$ rabbit normal IgG (#2729; Cell Signaling) and incubated at 4 °C overnight. The magnetic beads were

washed three times with buffer containing physiologic salt and once with low salt buffer. Then, the magnetic beads were treated with proteinase K at 65 °C for 2 h, and heated at 95 °C for 15 min for inactivation of proteinase K and elution of bound chromatin DNA. Eluted DNA and input chromatin DNA were analysed with a CFX Connect real-time PCR detection system (BioRad) using the KOD SYBR qPCR mix (TOYOBO) and the following primers: mouse *Il31* promoter (− 249/− 97) 5′-ATCTTCTGCCTTGCCTTGAG-3′ and 5′-ATGAGGAAGCA-GATCAAAGG-3′. Data were normalized for the amount of the input chromatin DNA.

**Statistical analysis.** Sample sizes for each experiment were determined on the variability observed in preliminary experiments and prior experience with the experimental systems. Statistical analyses were performed using GraphPad Prism. We first calculated the Gaussian distribution of the data using the Kolmogorov-Smirnov test. When two groups were compared, two-tailed Student's *t*-test (Gaussian distribution) or Mann–Whitney test (no Gaussian distribution) was used. When several groups were compared, we used a one-way analysis of variance (multiple groups) followed by *post hoc* Bonferroni test. *P* values less than 0.05 were considered significant.

**Data availability.** Microarray data that support the findings of this study have been deposited in Gene Expression Omnibus with the primary accession code GSE75686. The authors declare that all other data supporting the findings of this study are available within the article and its Supplementary Information Files.

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

## Acknowledgements

We thank T. Saito for AND TCR Tg mice; S. Yamasaki for Tg mice expressing Cre-recombinase under CD4 enhancer/promoter (CD4-Cre Tg mice); A. Miyajima and K. Nakamura for their kind permission to use $Osmr^{-/-}$ mice in this study; and K. Takeda and T. Akagi for plasmid vectors. We also thank A. Inayoshi, Siqinbala and A. Aosaka for technical assistance. This research is supported by the Leading Advanced Projects for Medical Innovation (LEAP; to Y.F.) and the Research on Development of New Drugs (to Y.F.) from Japan Agency for Medical Research and Development (AMED); and Grants-in-Aid for Scientific Research from the Ministry of Education, Culture, Sports, Science and Technology of Japan (to Y.F.) and the Japan Society for the Promotion of Science (to Y.F.).

## Author contributions

K.Y., T.U., A.S., Y.T., M.U. and M.W. performed functional, histological and biochemical analyses; K.Y., T.N., M.K.-N., I.T. and M.F. collected patient samples and analysed them; K.Y., T.U., M.F. and Y.F. contributed to writing the manuscript; Y.F. conceived the project, interpreted the data and wrote the manuscript.

## Additional information

**Competing financial interests:** The authors declare no competing financial interests.

