## [Peer Review File · Nature Communications]

Reviewers' comments:

Reviewer #1 (Remarks to the Author):

In this manuscript, Yamamura et al investigate an important problem: why DOCK8-deficiency causes atopic dermatitis and what this can tell us more generally about how atopic dermatitis develops in the general population. Using Dock8-deficient mice and genetic manipulations of atopic dermatitis patient cells, the authors delineate an EPAS1-/SP1- dependent pathway that increases transcription of IL-31 in CD4 T cells, leading to skin inflammation that is characteristic of atopic dermatitis. EPAS1 appears to have limited expression in the immune system, as least according to publically available microarray data. Thus, the identification of EPAS1 as a key transcriptional regulator of IL-31 that is downstream of DOCK8 is novel. The authors go on to define molecular interactions of DOCK8 with EPAS1 and MST1 that could regulate nuclear translocation of EPAS1 for IL-31 induction. Furthermore, no model of Dock8-deficient mice has been previously established that fully recapitulates the hyper-IgE phenotype found in the patients, so the data supporting a weaker TCR strength-of-signal in the disease pathogenesis is highly informative. The article is well written and appropriately references previous work.

Although their findings are very interesting, of definite physiological relevance for a widespread clinical problem (i.e., atopic dermatitis), and advance the field, several issues need attention, as described below:

Major points:

1. In general, the figures require improvements in their presentation and statistics to ensure reproducibility and strengthen the conclusions. The authors should present the mean with variability across the independent experiments, not a single representative experiment with variability based upon replicates in that single shown experiment (Fig 1a, 1b, 1e, 3c). The numbers of mice used for each group in each experiment should be clearly indicated. This information is needed to enable the reader to judge the strength of the data. Right now I have some concerns about whether limited mice/experiments were used/performed in some of the panels where one-tailed t-tests and SEM were displayed.
2. The authors mention that the mice were housed under specific pathogen-free conditions, but it is known that colonization with certain microbiota (*Staphylococcus aureus*, *Corynebacterium* species) can contribute to the development of atopic dermatitis (Kobayashi et al, *Immunity* 42:756, 2015). The authors should exclude these as a co-factor by microbiological testing or antibiotic treatment of their Dock8-/- mouse models.
3. The authors mapped the EPAS1-/SP-1 binding sites to the IL-31 promoter, but this was demonstrated in MEF using transfected constructs. Whether this actually happens endogenously in the relevant cell type hasn't yet been determined using ChIP assay in the activated pathogenic T cells. This should be done.
4. For the EPAS1 nuclear translocation studies in Fig 5a, I am confused because per the literature DOCK8 is not normally expressed in human fibroblasts. If this is also the case for MEF, then there shouldn't be any difference in EPAS1 localization in WT versus Dock8-/- MEF, unless there are other mutations in their mice not related to Dock8. Regarding the latter point, increasingly there is realization that mouse strains can have multiple mutations, for example *Nlrp10*-deficient mice later found to have Dock8 mutations. Can the authors clarify this point of confusion by performing western blotting for Dock8 in these cells? Related to this point, the authors haven't included any detailed information in the Methods regarding how many cells per experiment were quantitated - presumably the average % of cells for each experiment were used for the statistics shown in the right panel of Fig 5a?
5. The authors' model of disease pathogenesis is that MST1 and EPAS1 bind to the N-terminus of

DOCK8 to influence the nuclear localization of EPAS1. Presumably binding to DOCK8 is what keeps the MST1-EPAS1 complex within the cytosol. This conclusion is based upon co-immunoprecipitations of overexpressed constructs, the ability of overexpressed WT but not N-terminally deleted DOCK8 can suppress EPAS1 nuclear translocation in MEFs, and the ability of Mst1 knockdowns to impair nuclear translocation of EPAS1 and IL-31 expression. To provide further support for this model, the authors should demonstrate these interactions occur endogenously in the activated CD4 T cells.

Minor points:

1. The Methods section lacks detailed information regarding the sources of MEF, the adoptive transfer experiments, and the definition of a scratching bout (minimal time interval between bouts, minimal duration of a bout).
2. Although the pictures of the mice appear very impressive, quantitation of the extent of atopic dermatitis disease over time is needed to establish that these images are representative. I suggest use of the SCORAD (human SCORing Atopic Dermatitis), which has been modified for use in dogs and mice.
3. Do the OVA-transgenic Dock8^{-/-} mice have increased serum IgE like the AND TCR transgenic Dock8^{-/-} mice? It is implied that the decreased strength of signal through the AND TCR is responsible for the spontaneous skin inflammation and hyper-IgE that occur (besides the increased IL-31) in this mouse strain.
4. Microarray data should be included in the supplementary figures.
5. What was the efficiency of DOCK8 overexpression in HEK293T cells, given the protein's large size? Western blots should accompany the experiments in Fig 5b.

Reviewer #2 (Remarks to the Author):

Yamamura et al investigated the transcription factor EPAS1 that links DOCK8 deficiency to skin inflammation via IL-31 induction.

While I find the study interesting, I also have some concerns.

Major comments

The authors conclude that EPAS1 links DOCK8 deficiency to atopic skin inflammation via IL-31 induction in CD4⁺T cells. However, the focus on atopic dermatitis is completely unclear, since Hyper IgE-Syndrom has nothing to do with atopic dermatitis and nothing to do with atopy. Thus, the study is based on a wrong approach. Atopic dermatitis is a completely different disease and the authors must either focus on atopic dermatitis or on HIES. Hyper IgE-Syndrome is classified into Stat-3 deficient subtype and the more rare cases of DOCK8 deficiency with viral infections. The skin manifestations within these subtypes do not depend on atopy and this must be clarified in the manuscript presented. There is also a discrepancy between the title and the abstract. The abstract focuses on hyper IgE Syndrom and the title on atopic skin inflammation. Since the authors did not investigate atopic mice this must be changed. Thus, the authors need to clarify on which disease they like to focus on. Otherwise this is a mixture based on a wrong clinical attempt.

The CD4⁺ T cell count in DOCK8-HIES patients is lower than in patients with atopic dermatitis (Boos et al 2014, Allergy). Are there any data about the concentration of IL-31 in the serum of DOCK8 HIES patients in comparison to patients with atopic dermatitis? Is there any relation between CD4⁺ T cells and IL-31 concentrations in HIES?

As there are no known mutations in DOCK8 in patients with atopic dermatitis, DOCK8 can act as the natural negative regulator and the release of IL-31 of CD4⁺ T cells should be lower than in

DOCK8-HIES patients. Can the authors provide some information?

In case CD4+ T cells from patients with atopic dermatitis express less IL-31, which cells could be the IL-31 source? How cell type specific is the investigated signal cascade? Could it be possible that other cells express IL-31 independent of presented pathway?

If CD4+ T cells express higher amounts of IL-31 than DOCK8-HIES patients, how can this be explained?

Patient characterization is completely missing. Informed consent and ethics approval for samples derived from patients with atopic dermatitis is missing.

Minor comments

Since IL-31 is the most interesting gene, it should be mentioned in the written results part as first analysed gene, similar to Figure 1c. For a better understanding, the corresponding text should fit with the figures.

It is not clear why the authors used different concentrations of CD3 and at which time point they analysed the IL31 release (Fig. 1e).

In the paragraph "Atopic skin inflammation..." it is written: "... with massive infiltration of CD4+ T cells and, to lesser extent, eosinophils (Fig. 2 e, f). But in the Figure 2 total immune cells and eosinophils are shown as first results and then CD4+ T cells. The order should be changed to avoid

confusions, the label of the single parts of Figure 2 should be Dock8+/- AND Tg and/or Dock8-/- AND Tg. y-axis of Fig. 2e and 2i should be labeled with "total inflammatory cells/ 0.25mm²" / "eosinophils/ 0.25mm²"

In the paragraph "Identification of EPAS1 as a master regulator...", it is not clear that WT CD4+ T Cells were firstly stimulated and then transduced with the retrovirus.

There is no information about the stimulation of the CD4+T cells in Fig. 3c? Is this the same concentration as used for the secondary stimulation (Methods page 16)

Labeling of the y-axis of Fig 3f: total inflammatory cells / 0.25mm²

It would be interesting to get some more information about the finding of the DOCK8 binding protein MTS1?

Description of the analysis of the IL-31 promoter should be revised in relation to the figure. In Figure 4a, there are four mutants illustrated, but in the paragraph only two of them are mentioned and in the legend there is also no further explanation. The explanation of the results is not in line with the Figure.

How was the promoter activity (Fig. 4a, b) calculated? Is the promoter activity present in relation to mock?

There are Dock8-/- (Fig. 5a) and Epas1-/- MEF (Fig. 4f) used, but they are not mention in Material and Methods. Please provide information about these cells.

In the legends should be mention which types of cells are used for the experiments.

In general it is not clear what is shown in the agarosegels of Fig. 3b, 5e, 6a, b. These are two different donors or different concentration of one PCR sample?

It is not clear if the authors established for the experiments in Fig. 5c a stable HEK cell line, which overexpress the proteins continuously, or is this only a transient transfection. Regarding to this it would be good to know at which time point after transfection the authors analysed the cells.

As it is not mention in the legend of Fig 6a, it is unclear if the expression was calculated in

comparison to unstimulated or not? And what was the stimulation?

Fig. 6b: How long were the CD4+ T cells cultivated and did the authors use a stimulus?

The information about the PCR is confusing. What does RT-PCR mean? Accordingly to the description it should be named as a conventional PCR. In case the RT is used as the abbreviation for reverse transcriptase, this is wrong in this context, as the authors described the reverse transcription as an own step.

Page 21 the centrifugal force is noted as 2000rpm, please provide the information as g.

Please provide some more information about the antibodies, at least the clone or the catalog number.

In the statistical part it should be mentioned which test was used to analyse the Gaussian distribution. Further the statistical analysis should be revised. The analysis in figure 1 was only performed with Student's t-Test, but for example Fig 1c should be analysed with one-way ANOVA, as there are two groups with more than two variables.

In the introduction the reference should be revised (ref 3 - 5).

Since the authors investigated the role of CD4+ T cells and their expression of IL-31 under the influence of DOCK8, which is involved in IES, some information about the T cells counts in wildtype mice compared to the Dock8^{-/-} mice would be good.

Related to the reference 1. Minegishi Y. & Saito M. it would be better to write: Hyper IgE syndrome (HIES) is a primary immunodeficiency characterized by atopic dermatitis-like skin.

In general for the manuscript, the gene symbols should be revised. It should be always IL-31/ IL-31 or IL31/ IL31 and OSMR^{-/-} should be written Osmr^{-/-} for mice

Reviewer #3 (Remarks to the Author):

Yamamura et al study the etiology for the atopic phenotype observed in DOCK8 deficiency, since no clear evidence has yet been presented for why this immune deficiency leads to such a robust set of allergic diseases. They find elevated IL-31 in the skin of DOCK8 deficient mice, and are able to induce AD in these mice by breeding OTII to an OVA producing mouse, or by breeding it to the AND strain which leads to the observed atopic dermatitis. Mechanistically, they show that DOCK8, along with MST1 normally binds to EPAS1 and negatively regulates EPAS1 translocation to the nucleus in CD4+ T-cells. EPAS1, when disinhibited, up regulates IL-31 via SP1 in CD4+ T-cells. The IL-31 activity acts via OSMR, since DOCK8/OSMR^{-/-} mice do not develop severe dermatitis.

1. It is worth noting that some MST1 mutant patients were reported to have modest IgE elevation, not just the one patient cited in the discussion. It should be clarified as to whether those patients have atopic dermatitis, since the data presented suggest AD would be the prime driver of the atopic phenotype.
2. Along those lines, is there a particular reason neither MST1 nor DOCK8^{mut} patient CD4 cells were not used in this study? Access to these patient samples should not be difficult, and their study is a rather key element to demonstrating the human relevance of the findings presented. The atopic dermatitis
3. The authors show that DOCK8 appears to be important for preventing EPAS1 transcription and

for sequestering EPAS1 in the cytoplasm-- is either the dominant regulating element? EPAS1 appears to promote its own transcription as well, based on figure 6. This regulatory loop could use clarification, perhaps in the discussion.

4. What is the reason typical AD patients would poor control over EPAS1? Is dock8 deficient? Are the complexes not formed as well?

Minor point: There are multiple syndromes causing elevated IgE, and there are far more cases of "hyper-IgE" with no syndromic comorbidities, infections, evidence of immune deficiency or genetic cause at all. It is therefore inaccurate to describe HIES as a primary immunodeficiency. If the authors wish, they could mention that STAT3LOF is often called the autosomal dominant hyper-IgE syndrome. DOCK8 deficiency is an example of a syndrome which can be associated with marked IgE elevation.

We found the reviewers' comments most helpful and have revised the manuscript accordingly. Our responses to the reviewers' comments are as follows:

<To the Reviewer 1>

1) In general, the figures require improvements in their presentation and statistics to ensure reproducibility and strengthen the conclusions. The authors should present the mean with variability across the independent experiments, not a single representative experiment with variability based upon replicates in that single shown experiment (Fig 1a, 1b, 1e, 3c). The numbers of mice used for each group in each experiment should be clearly indicated. This information is needed to enable the reader to judge the strength of the data. Right now I have some concerns about whether limited mice/experiments were used/performed in some of the panels where one-tailed t-tests and SEM were displayed.

According to your advice, we have shown the mean with variability (mean \pm s.d.) of multiple samples obtained from independent experiments in all figures including Fig 1b, 1c, 1f, and 3c. We have also indicated the number of mice used for each group in each experiment. In addition, we have increased the number of mice or experiments to strengthen the data in some figures.

2. The authors mention that the mice were housed under specific pathogen-free conditions, but it is known that colonization with certain microbiota (*Staphylococcus aureus*, *Corynebacterium* species) can contribute to the development of atopic dermatitis (Kobayashi et al, *Immunity* 42:756, 2015). The authors should exclude these as a co-factor by microbiological testing or antibiotic treatment of their *Dock8*^{-/-} mouse models.

As you pointed out, it has been reported that colonization of *Staphylococcus aureus* and *Corynebacterium bovis* contribute to the development of atopic dermatitis (AD). Therefore, it is possible that dysbiosis in the skin would promote disease development in *Dock8*^{-/-} AND Tg mice. To address this possibility, we prepared facial skin swab samples from *Dock8*^{+/-} and *Dock8*^{-/-} AND or OTII Tg mice at 6 weeks old, and tried to detect *Staphylococcus aureus* by culture or *Corynebacterium bovis* by PCR. As a result, neither *Staphylococcus aureus* nor *Corynebacterium bovis* was detected in our hands (please see the data in the next page). Therefore, we believe that AD-like skin inflammation would occur in *Dock8*^{-/-} AND Tg mice, independently of colonization of these bacteria. However, since these experiments are still preliminary, we do not refer to this

possibility in the revised manuscript.

Facial skin swab samples were obtained from *Dock8*^{+/-} and *Dock8*^{-/-} AND or OTII Tg mice at 6 weeks old.

(a) Samples were placed into 1 ml of PBS, 100 μ l of which was plated on Mannitol salt agar with egg yolk to detect *Staphylococcus aureus*, as previously described (Kobayashi T. et al., *Immunity* 42: 756-766, 2015). (b) Genome DNA was isolated from facial skin swab samples and was subjected to PCR for detection of *Corynebacterium bovis* with the following primers: *C. bovis*; 5'-GGTGTGGGGATCTTCCACGAT-3' and 5'-ACCACCTGTGAACAAGCCCA-3'. *16S rRNA*; 5'-AGAGTTTGATCCTGGCTCAG-3' and 5'-GACGGGCGGTGTGTRCA-3' (Duga S. et al., *Molecular and Cellular probes* 12: 191-199, 1998).

3) The authors mapped the EPAS1-/SP-1 binding sites to the IL-31 promoter, but this was demonstrated in MEF using transfected constructs. Whether this actually happens endogenously in the relevant cell type hasn't yet been determined using ChIP assay in the activated pathogenic T cells. This should be done.

According to your advice, we have performed ChIP assay using CD4⁺ T cells from CD4-Cre⁻ or CD4-Cre⁺ *Epas1*^{lox/lox} *Dock8*^{-/-} AND Tg mice. In the revised manuscript, these results are shown in Figure 4f, and described as follows: **'In addition, chromatin immunoprecipitation (ChIP) assay revealed that SP1 was recruited to *Il31* promoter in activated CD4⁺ T cells from *Dock8*^{-/-} AND Tg mice and its recruitment was significantly reduced in the absence of EPAS1 (Fig. 4f).'**

4) For the EPAS1 nuclear translocation studies in Fig 5a, I am confused because per the literature DOCK8 is not normally expressed in human fibroblasts. If this is also the case for MEF, then there shouldn't be any difference in EPAS1 localization in WT versus *Dock8*^{-/-} MEF, unless there are other mutations in their mice

not related to Dock8. Regarding the latter point, increasingly there is realization that mouse strains can have multiple mutations, for example Nlrp10-deficient mice later found to have Dock8 mutations. Can the authors clarify this point of confusion by performing western blotting for Dock8 in these cells?

DOCK8 was initially identified as a gene, the expression level of which is reduced in lung cancers (Takahashi K. et al. International Journal of Oncology 28: 321-328, 2006). Indeed, *DOCK8* is expressed in various tissues and various cell-types, although the expression levels are variable among cells. In the revised manuscript, we have provided Western blot data showing that *DOCK8* is expressed in MEFs (Supplementary Figure 6) and described as follows: ‘**As *DOCK8* is also expressed in MEFs (Supplementary Fig. 6), we prepared WT and *Dock8*^{-/-} MEFs, and examined the effect of *DOCK8* deficiency on subcellular localization of EPAS1 by staining them with anti-EPAS1 antibody.**’

5) The authors haven't included any detailed information in the Methods regarding how many cells per experiment were quantitated - presumably the average % of cells for each experiment were used for the statistics shown in the right panel of Fig 5a?

As you pointed out, we have analysed 30 cells per group in each experiment and used the average % of cells for the statistics. In the revised manuscript, this information has been provided in the legends to Figure 5a, 5c and 5f.

6) The authors' model of disease pathogenesis is that MST1 and EPAS1 bind to the N-terminus of DOCK8 to influence the nuclear localization of EPAS1. Presumably binding to DOCK8 is what keeps the MST1-EPAS1 complex within the cytosol. This conclusion is based upon co-immunoprecipitations of overexpressed constructs, the ability of overexpressed WT but not N-terminally deleted DOCK8 can suppress EPAS1 nuclear translocation in MEFs, and the ability of Mst1 knockdowns to impair nuclear translocation of EPAS1 and IL-31 expression. To provide further support for this model, the authors should demonstrate these interactions occur endogenously in the activated CD4 T cells.

According to your advice, we have analysed the ‘endogenous’ association between *DOCK8* and MST1 in the activated CD4⁺ T cells. In the revised manuscript, these results are shown in Figure 5e, and described as follows: ‘**During the course of screening for *DOCK8*–binding proteins, we found that *DOCK8* bound to MST1 through the N-terminal region when overexpressed in human embryonic kidney 293T (HEK-293T) cells (Fig. 5d, left). Similar association was observed with the activated CD4⁺ T cells (Fig. 5e).**’

7) The Methods section lacks detailed information regarding the sources of MEF, the adoptive transfer experiments, and the definition of a scratching bout (minimal time interval between bouts, minimal duration of a bout).

According to your comment, we have added the detailed information on MEFs, the adoptive transfer experiments, and the definition of a scratching bout in the Method section of the revised manuscript.

8) Although the pictures of the mice appear very impressive, quantitation of the extent of atopic dermatitis disease over time is needed to establish that these images are representative. I suggest use of the SCORAD (human SCORing Atopic Dermatitis), which has been modified for use in dogs and mice.

According to your advice, we have used SCORAD index to evaluate the severity of skin disease. In the revised manuscript, these results are shown in Figure 2b and described as follows: **‘The severity of dermatitis was grossly assessed by the SCORAD (human SCORing Atopic Dermatitis) that was modified for the use in mice. Irrespective of the sex of mice, skin phenotype first appeared 7–8 weeks after birth, generally worsened with the age, and was never observed in *Dock8*^{+/-} AND Tg littermate mice (Fig. 2b).’**

9) Do the OVA-transgenic *Dock8*^{-/-} mice have increased serum IgE like the AND TCR transgenic *Dock8*^{-/-} mice? It is implied that the decreased strength of signal through the AND TCR is responsible for the spontaneous skin inflammation and hyper-IgE that occur (besides the increased IL-31) in this mouse strain.

We have compared serum IgE levels between *Dock8*^{+/-} and *Dock8*^{-/-} OTII Tg mice. In the revised manuscript, these data are shown in Supplementary Figure 1, and described as follows: **‘Although IL31 has been implicated in pruritus in AD, *Dock8*^{-/-} OTII Tg mice showed neither skin inflammation nor IgE elevation (Supplementary Fig. 1).’**

10) Microarray data should be included in the supplementary figures.

According to your advice, we have included in the revised manuscript the microarray data in Supplementary Table 1.

11) What was the efficiency of DOCK8 overexpression in HEK293T cells, given the protein's large size? Western blots should accompany the experiments in Fig 5b.

We have used *Dock8*^{-/-} MEFs stably expressing HA-tagged WT DOCK8 or its mutants (Δ DHR2 and Δ N). To clarify this issue, we have provided Western blot data in Figure 5b, and described as follows: ‘**This effect of DOCK8 deficiency was cancelled when WT DOCK8 was stably expressed in *Dock8*^{-/-} MEFs (Fig. 5b,c). Similar results were obtained by expressing the DOCK8 mutant (Δ DHR2) lacking DOCK homology region (DHR)-2 domain critical for Cdc42 activation (Fig. 5b,c).**’

<To the Reviewer 2>

1) The authors conclude that EPAS1 links DOCK8 deficiency to atopic skin inflammation via IL-31 induction in CD4+T cells. However, the focus on atopic dermatitis is completely unclear, since Hyper IgE-Syndrom has nothing to do with atopic dermatitis and nothing to do with atopy. Thus, the study is based on a wrong approach. Atopic dermatitis is a completely different disease and the authors must either focus on atopic dermatitis or on HIES.

Previous studies have indicated ‘atopic dermatitis (AD)’ as a major clinical feature of DOCK8 immunodeficiency syndrome (Zhang Q. et al. Disease Markers 29: 131–139, 2010; Freeman A. and Holland S.M.), as you can see below.

Manifestations	Zhang et al. [1]	Engelhardt et al. [2]	
Atopic dermatitis	100%	95%	
Allergies	82%	48%	
Skin and soft tissue infections	82%	81%	
Sinopulmonary infections	91%	100%	
Candida infections	45%	81%	
Cutaneous viral infections	100%	71%	
	Herpes Simplex Virus	64%	48%
	Human Papilloma Virus	64%	14%
	Molluscum Contagiosum	45%	33%
	Others	36%	14%
Malignancy	36%	10%	

Two patients were reported in both papers (Patient 8-1 and Patient 8-2 as ARH11.4 and ARH11.5, respectively). These patients were included in each of the calculations for both series [1,2].

Therefore, we have used the term of ‘AD’ as a clinical feature associated with DOCK8 deficiency. However, this usage of AD may not be appropriate in our paper, as you have pointed out. Therefore, in the revised manuscript, we have changed all descriptions of ‘AD’ to ‘AD-like skin disease (inflammation)’ or just ‘skin inflammation’.

2) Hyper IgE-Syndrome is classified into Stat-3 deficient subtype and the more rare cases of DOCK8 deficiency with viral infections. The skin manifestations within these subtypes do not depend on atopy and

this must be clarified in the manuscript presented.

In the revised manuscript, we have referred to viral infection and described as follows: ‘**Homozygous and compound heterozygous mutations in *DOCK8* cause a combined immunodeficiency characterized by recurrent viral infections, early onset malignancy and atopic dermatitis (AD)-like skin lesion.**’

3) There is also a discrepancy between the title and the abstract. The abstract focuses on hyper IgE Syndrom and the title on atopic skin inflammation. Since the authors did not investigate atopic mice this must be changed. Thus, the authors need to clarify on which disease they like to focus on.

In the revised manuscript, we have changed the title as follows: ‘**The transcription factor EPAS1 links DOCK8 deficiency to skin inflammation via IL31 induction**’.

4) The CD4⁺ T cell count in DOCK8-HIES patients is lower than in patients with atopic dermatitis (Boos et al 2014, Allergy). Are there any data about the concentration of IL-31 in the serum of DOCK8 HIES patients in comparison to patients with atopic dermatitis? Is there any relation between CD4⁺ T cells and IL-31 concentrations in HIES?

As you pointed out, the number of CD4⁺ T cells was reduced in *Dock8*^{-/-} mice. In the revised manuscript, we have shown these data in Figure 1a,b and Supplementary Figure 2a,b. However, these CD4⁺ T cells produce large amounts of IL31 upon stimulation, and *Dock8*^{-/-} AND Tg mice exhibit increased levels of serum IL31. In addition, we have shown in the revised manuscript that the serum IgE level markedly increased in a DOCK8-deficient patient, as compared with those of healthy controls.

5) As there are no known mutations in DOCK8 in patients with atopic dermatitis, DOCK8 can act as the natural negative regulator and the release of IL-31 of CD4⁺ T cells should be lower than in DOCK8-HIES patients. Can the authors provide some information? In case CD4⁺ T cells from patients with atopic dermatitis express less IL-31, which cells could be the IL-31 source? How cell type specific is the investigated signal cascade? Could it be possible that other cells express IL-31 independent of presented pathway? If CD4⁺ T cells express higher amounts of IL-31 than DOCK8-HIES patients, how can this be explained?

As far as we have analysed the expression of DOCK8 in healthy controls and AD patients, no significant difference was found (Supplementary Figure 8). However, genetic and pharmacological inactivation of EPAS1 markedly suppressed IL31 production by CD4⁺ T cells from AD patients (Figure 6d–f). These results

suggest that DOCK8-independent, but EPAS-dependent pathway operates for IL31 induction in CD4⁺ T cells from AD patients. Therefore, in the revised manuscript, we have discussed as follows: **‘How EPAS1 is activated in CD4⁺ T cells from AD patients is currently known. However, recent evidence indicates that in human CD4⁺ T cells, EPAS1 is a direct target of STAT6 and serves as a hub protein in IL4-mediated transcription circuitries. Therefore, it is highly conceivable that multiple genetic and environmental factors skewing Th2 differentiation could contribute to EPAS1 activation in AD patients. Whether particular cascades are involved in EPAS1 activation in AD patients would be an important issue that should be investigated in future studies.’** On the other hand, we were unable to analyse CD4⁺ T cells from a DOCK8-deficient patient, because fortunately she had received bone marrow transplantation.

6) Patient characterization is completely missing. Informed consent and ethics approval for samples derived from patients with atopic dermatitis is missing.

We have described in the Method section (Preparation and activation of human CD4⁺ T cells) as follows: **‘The experiments were approved by the Ethics committee of Kyushu University Hospital and Fujita Health University, and a written informed consent was obtained from all patients.’**

7) Since IL-31 is the most interesting gene, it should be mentioned in the written results part as first analysed gene, similar to Figure 1c. For a better understanding, the corresponding text should fit with the figures. It is not clear why the authors used different concentrations of CD3 and at which time point they analysed the IL31 release (Fig. 1e).

According to your advice, we have described in the revised manuscript as follows: **‘Interestingly, however, we found that the level of *Il31* transcript markedly increased in CD4⁺ T cells from *Dock8*^{-/-} OTII Tg mice 24 h after stimulation, as compared with that in *Dock8*^{+/-} OTII Tg CD4⁺ T cells (Fig. 1d). In contrast, *Il2* and *Il4* gene expressions in stimulated CD4⁺ T cells were unchanged between *Dock8*^{+/-} and *Dock8*^{-/-} OTII Tg mice (Fig. 1d).’**

We have used different concentrations of anti-CD3e antibody to see whether IL31 induction occurs through a mechanism dependent on TCR signals.

8) In the paragraph "Atopic skin inflammation..." it is written: "... with massive infiltration of CD4+ T cells and, to lesser extent, eosinophils (Fig. 2 e, f). But in the Figure 2 total immune cells and eosinophils are shown as first results and then CD4+ T cells. The order should be changed to avoid confusions, the label of the single parts of Figure 2 should be Dock8+/- AND Tg and/or Dock8-/- AND Tg. y-axis of Fig. 2e and 2i should be

labeled with "total inflammatory cells/ 0.25mm.

According to your advice, we have labeled Figure 2f, 2j, and 3f with total inflammatory cells/ mm². In addition, to avoid confusion, we have modified Figure 2f and shown the data on eosinophils in Supplementary Figure 3.

9) In the paragraph "Identification of EPAS1 as a master regulator...", it is not clear that WT CD4⁺ T Cells were firstly stimulated and then transduced with the retrovirus.

For retroviral gene expression, CD4⁺ T cells must be activated beforehand. In addition, it is also conceivable that activated status is required to induce epigenetic modification in CD4⁺ T cells for IL31 induction.

10) There is no information about the stimulation of the CD4⁺T cells in Fig. 3c? Is this the same concentration as used for the secondary stimulation (Methods page 16).

To clarify this issue, we have described in the legend as follows: ‘**Effect of genetic inactivation of *Epas1* on *Ii31* gene expression in CD4⁺ T cells from *Dock8*^{-/-} AND Tg mice after primary stimulation with MCC peptide.**’

11) It would be interesting to get some more information about the finding of the DOCK8 binding protein MTS1?

According to your advice, we have described in the discussion section as follows: ‘**Interestingly, it has been reported that 7 of 9 patients with *MST1* mutations had eczema or AD-like skin disease, raising the possibility that *MST1* is also involved in IL31 induction.**’

12) Description of the analysis of the IL-31 promoter should be revised in relation to the figure. In Figure 4a, there are four mutants illustrated, but in the paragraph only two of them are mentioned and in the legend there is also no further explanation. The explanation of the results is not in line with the Figure.

According to your advice, we have described in the revised manuscript as follows: ‘**When this reporter construct was expressed in mouse embryonic fibroblasts (MEFs), *Ii31* promoter activation was induced in the presence of WT EPAS1 (Fig. 4a). However, the expression of EPAS1 mutants lacking N-terminal or C-terminal activation domain (Δ N-TAD and Δ C-TAD) failed to induce promoter activation (Fig. 4a).**

EPAS1 is known to form a complex with ARNT via PAS-B domain. Surprisingly, however, deletion of neither PAS-B nor bHLH did affect *I131* promoter activation (Fig. 4a). Consistent with this, EPAS1-mediated *I131* promoter activation occurred normally even when *Arnt* gene was knocked down (Fig. 4b).’

13) How was the promoter activity (Fig. 4a, b) calculated? Is the promoter activity present in relation to mock?

To clarify this issue, we have described in the legend in the revised manuscript as follows: ‘**In (a-d), promoter activity is expressed as the relative index after normalization of the luciferase activity of pGL4.10-*I131* alone (Mock) to an arbitrary value of 1.**’

14) There are *Dock8*^{-/-} (Fig. 5a) and *Epas1*^{-/-} MEF (Fig. 4f) used, but they are not mention in Material and Methods. Please provide information about these cells.

In the revised manuscript, we have provided information on these MEFs in the Method section.

15) In the legends should be mention which types of cells are used for the experiments.

In the revised manuscript, we have described in the legend or text which types of cells were used for the experiments.

16) In general it is not clear what is shown in the agarosegels of Fig. 3b, 5e, 6a, b. These are two different donors or different concentration of one PCR sample?

To clarify this issue, we have described in the legends as follows: ‘**Amplification increased by 2 cycles, from the left to the right, starting at 34 cycles for *Epas1* and *I131*, or at 29 cycles for *Gapdh*** (for Figure 3b).’

17) It is not clear if the authors established for the experiments in Fig. 5c a stable HEK cell line, which overexpress the proteins continuously, or is this only a transient transfection. Regarding to this it would be good to know at which time point after transfection the authors analysed the cells.

We have used *Dock8*^{-/-} MEFs stably expressing HA-tagged WT DOCK8 or its mutants (Δ DHR2 and Δ N). To clarify this issue, we have provided Western blot data in Figure 5b, and described as follows: ‘**This effect of**

DOCK8 deficiency was cancelled when WT DOCK8 was stably expressed in *Dock8*^{-/-} MEFs (Fig. 5b,c). Similar results were obtained by expressing the DOCK8 mutant (Δ DHR2) lacking DOCK homology region (DHR)-2 domain critical for Cdc42 activation (Fig. 5b,c).'

18) As it is not mention in the legend of Fig 6a, it is unclear if the expression was calculated in comparison to unstimulated or not? And what was the stimulation?

To clarify this issue, we have described in the legend in the revised manuscript as follows: '**Expression (fold increase) is relative to that of the unstimulated samples.'**

19) Fig. 6b: How long were the CD4+ T cells cultivated and did the authors used a stimulus?

The data on Figure 6b shows serum concentration of IL31 in a DOCK8-deficient patient, AD patients and healthy controls.

20) The information about the PCR is confusing. What does mean RT-PCR means? Accordingly to the description it should be named as a conventional PCR.

According to your comment, we have described 'conventional RT-PCR' in the Method section.

21) Page 21 the centrifugal force is noted as 2000rpm, please provide the information as g.

In the revised manuscript, we have described as 700g.

22) Please provide some more information about the antibodies, at least the clone or the catalog number.

In the revised manuscript, we have provided additional information on antibodies.

23) In the statistical part it should be mentioned which test was used to analyse the Gaussian distribution. Further the statistical analysis should be revised. The analysis in figure 1 was only performed with Student's t-Test, but for example Fig 1c should be analysed with one-way ANOVA, as there are two groups with more than two variables.

According to your advice, we have re-analysed all data and described in the Method section as follows:

‘Statistical analyses were performed using GraphPad Prism. We first calculated the Gaussian distribution of the data using the Kolmogorov-Smirnov test. When two groups were compared, two-tailed Student’s *t*-test (Gaussian distribution) or Mann-Whitney test (no Gaussian distribution) was used. When several groups were compared, we used a one-way ANOVA (multiple groups) followed by *post hoc* Bonferroni test. *P* values less than 0.05 were considered significant.’ On the other hand, when we compared between two groups under the same condition in multiple panels, we have described in the revised manuscript as follows: ‘**Comparison was made between two groups at the same time point** (for Fig. 1d).’

24) In the introduction the reference should be revised (ref 3 - 5).

In the revised manuscript, we have changed Introduction section.

25) Since the authors investigated the role of CD4+ T cells and their expression of IL-31 under the influence of DOCK8, which is involved in IES, some information about the T cells counts in wildtype mice compared to the Dock8-/- mice would be good.

In the revised manuscript, we have shown these data in Figure 1a,b and Supplementary Figure 2a,b.

26) Related to the reference 1.Minegishi Y. & Saito M. it would be better to write: Hyper IgE syndrome (HIES) is a primary immunodeficiency characterized by atopic dermatitis-like skin.

According to your advice, we have changed all descriptions of ‘AD’ to ‘AD-like skin disease (inflammation)’ or just ‘skin inflammation’.

27) In general for the manuscript, the gene symbols should revised. It should be always IL-31/ Il-31 or IL31/ Il31 and OSMR-/- should be written Osmr-/- for mice.

According to your advice, we have revised the gene symbols throughout the text.

<To the Reviewer 3>

1) It is worth noting that some MST1 mutant patients were reported to have modest IgE elevation, not just the one patient cited in the discussion. It should be clarified as to whether those patients have atopic dermatitis,

since the data presented suggest AD would be the prime driver of the atopic phenotype.

As you pointed out, previous literatures have indicated that patients with MST1 mutations have eczema or AD-like skin lesion. Therefore, according to your advice, we have described in the Discussion section as follows: **‘Interestingly, it has been reported that 7 of 9 patients with *MST1* mutations had eczema or AD-like skin disease, raising the possibility that MST1 is also involved in IL31 induction. Indeed, we found that *IL31* gene expression markedly increased in CD4⁺ T cells when *Mst1* gene was knocked down.’**

2) Along those lines, is there a particular reason neither MST1 nor DOCK8mut patient CD4 cells were not used in this study? Access to these patient samples should not be difficult, and their study is a rather key element to demonstrating the human relevance of the findings presented.

As it takes long time to make ethical agreement internationally, we have analysed a DOCK8-deficient patient in Japan. In the revised manuscript, we have shown that this patient exhibited an elevated serum IL31 level, as seen in conventional AD patients. On the other hand, we were unable to analyse her CD4⁺ T cells, because fortunately she had received bone marrow transplantation. In future studies, we would like to collaborate with many clinicians and scientists all over the world to reveal the role of DOCK8, EPAS1 and/or MST1 in the disease pathogenesis.

3) The authors show that DOCK8 appears to be important for preventing EPAS1 transcription and for sequestering EPAS1 in the cytoplasm-- is either the dominant regulating element? EPAS1 appears to promote its own transcription as well, based on figure 6. This regulatory loop could use clarification, perhaps in the discussion.

I am very sorry to make confusion: FM19G11 is not a direct inhibitor of EPAS1, but inhibits EPAS1 gene expression by acting on ‘undefined’ target (Moreno-Manzano V. et al. J. Biol. Chem. 285: 1333–1342, 2010). To avoid confusion, we have described in the revised manuscript as follows: **‘To further examine the role of EPAS1 in IL31 induction, we treated CD4⁺ T cells from AD patients with two inhibitors, FM19G11 and HIFVII. FM19G11 is not a direct inhibitor of EPAS1, but it is known to suppress *EPAS1* gene expression by acting on undefined target. Indeed, the expression level of EPAS1 was markedly reduced when human CD4⁺ T cells were treated with FM19G11 (Fig. 6e). In agreement with genetic data, we found that FM19G11 treatment ablated TCR stimulation–induced expression of *IL31*, but not *IL2*, in CD4⁺ T cells from AD patients (Fig. 6f)’.**

4) What is the reason typical AD patients would poor control over EPAS1? Is dock8 deficient? Are the complexes not formed as well?

As far as we have analysed the expression of DOCK8 in healthy controls and AD patients, no significant difference was found (Supplementary Figure 8). However, genetic and pharmacological inactivation of EPAS1 markedly suppressed IL31 production by CD4⁺ T cells from AD patients (Figure 6d–f). These results suggest that DOCK8-independent, but EPAS-dependent pathway operates for IL31 induction in CD4⁺ T cells from AD patients. Therefore, in the revised manuscript, we have discussed as follows: **‘How EPAS1 is activated in CD4⁺ T cells from AD patients is currently known. However, recent evidence indicates that in human CD4⁺ T cells, EPAS1 is a direct target of STAT6 and serves as a hub protein in IL4-mediated transcription circuitries. Therefore, it is highly conceivable that multiple genetic and environmental factors skewing Th2 differentiation could contribute to EPAS1 activation in AD patients. Whether particular cascades are involved in EPAS1 activation in AD patients would be an important issue that should be investigated in future studies.’** Since many CD4⁺ T cells are required for immunoprecipitation, we were unable to examine whether the association between DOCK8 and EPAS1 normally occurs in CD4⁺ T cells from AD patients. As described in the Discussion section, we believe that this should be investigated in future studies.

5) There are multiple syndromes causing elevated IgE, and there are far more cases of "hyper-IgE" with no syndromic comorbidities, infections, evidence of immune deficiency or genetic cause at all. It is therefore inaccurate to describe HIES as a primary immunodeficiency. If the authors wish, they could mention that STAT3LOF is often called the autosomal dominant hyper-IgE syndrome. DOCK8 deficiency is an example of a syndrome which can be associated with marked IgE elevation.

According to your suggestion, we have removed the part of ‘hyper IgE’ and revised the Introduction section as follows: **‘Homozygous and compound heterozygous mutations in *DOCK8* cause a combined immunodeficiency characterized by recurrent viral infections, early onset malignancy and atopic dermatitis (AD)-like skin lesion. Since patients with *DOCK8* mutations exhibit markedly elevated serum IgE levels, this disorder has been classified as a form of autosomal recessive hyper IgE syndrome (HIES). *DOCK8* is an evolutionarily conserved guanine nucleotide exchange factor (GEF) for Cdc42. Accumulating evidence indicates that human patients with *DOCK8* mutations have morphological and functional abnormalities of leukocytes.’**

REVIEWERS' COMMENTS:

Reviewer #1 (Remarks to the Author):

In this revised manuscript, Yamamura et al have satisfactorily addressed the concerns I previously raised.

Overall, I think this is an excellent piece of work, well-written and now suitable for publication, which delineates the cellular and molecular mechanism behind the atopic dermatitis occurring in DOCK8-deficiency. The findings are novel and conclusions are well supported by the data presented.

Reviewer #2 (Remarks to the Author):

The authors have done an extensive revision. However, I am still missing the patients details for the patients with atopic Dermatitis and healthy controls.

Reviewer #3 (Remarks to the Author):

The authors have responded adequately to all of my concerns. It is worth noting that the dermatitis seen in DOCK8 deficiency is indistinguishable from typical severe atopic dermatitis and as such I see no reason to change the language to "AD-like". As written now in the revision, it would be inaccurate and a mischaracterization. STAT3LOF associated AD-HIES can lead to a dermatitis that might be distinct from typical AD (and perhaps that is the point of confusion), but STAT3LOF is not related to the disease studied here.

One other point-- The elevated IL-31 data from the serum of a transplanted DOCK8 patient creates confusion-- if the source of the IL-31 should be CD4 cells and there aren't any host CD4 cells left, then there are questions raised about the relevance of many of the findings.

We thank all reviewers for their careful reading of the manuscript. As the Reviewer 1 did not raise the additional issues, we have revised the manuscript according to the comments from the Reviewer 2 and 3. Our responses to the reviewers' comments are as follows:

<Reviewer 1>

In this revised manuscript, Yamamura et al have satisfactorily addressed the concerns I previously raised. Overall, I think this is an excellent piece of work, well-written and now suitable for publication, which delineates the cellular and molecular mechanism behind the atopic dermatitis occurring in DOCK8-deficiency. The findings are novel and conclusions are well supported by the data presented.

Thank you for your valuable comments during review process.

<Reviewer 2>

The authors have done an extensive revision. However, I am still missing the patients details for the patients with atopic Dermatitis and healthy controls.

According to your comment, we have included the demographic information of AD patients and healthy controls in Supplementary Table 2.

<Reviewer 3>

The authors have responded adequately to all of my concerns. It is worth noting that the dermatitis seen in DOCK8 deficiency is indistinguishable from typical severe atopic dermatitis and as such I see no reason to change the language to "AD-like". As written now in the revision, it would be inaccurate and a mischaracterization . STAT3LOF associated AD-HIES can lead to a dermatitis that might be distinct from typical AD (and perhaps that is the point of confusion), but STAT3LOF is not related to the disease studied here.

According to your advice, we have used 'atopic dermatitis' or 'atopic skin inflammation' throughout the text. Accordingly, we have also revised the title as follows: 'The transcription factor EPAS1 links DOCK8 deficiency to atopic skin inflammation via IL-31 induction'.

The elevated IL-31 data from the serum of a transplanted DOCK8 patient creates confusion-- if the source of the IL-31 should be CD4 cells and there aren't any host CD4 cells left, then there are questions raised about the relevance of many of the findings.

I am very sorry to make confusion. We have analyzed IL-31 concentration in the serum obtained from the patient before bone marrow transplantation. Therefore, the source of the IL-31 should be DOCK8-deficient CD4⁺ T cells.